# Strategic Littlestone Dimension:
# Improved Bounds on Online Strategic Classification*

**Saba Ahmadi**[†,*]**, Kunhe Yang**[‡,*]**, and Hanrui Zhang**[§,*]

[†]Toyota Technological Institute at Chicago, `saba@ttic.edu`
[‡]University of California, Berkeley, `kunheyang@berkeley.edu`
[§]Chinese University of Hong Kong, `hanrui@cse.cuhk.edu.hk`

## Abstract

We study the problem of online binary classification in settings where strategic agents can modify their observable features to receive a positive classification. We model the set of feasible manipulations by a directed graph over the feature space, and assume the learner only observes the manipulated features instead of the original ones. We introduce the *Strategic Littlestone Dimension*, a new combinatorial measure that captures the joint complexity of the hypothesis class and the manipulation graph. We demonstrate that it characterizes the instance-optimal mistake bounds for deterministic learning algorithms in the realizable setting. We also achieve improved regret in the agnostic setting by a refined agnostic-to-realizable reduction that accounts for the additional challenge of not observing agents' original features. Finally, we relax the assumption that the learner knows the manipulation graph, instead assuming their knowledge is captured by a family of graphs. We derive regret bounds in both the realizable setting where all agents manipulate according to the same graph within the graph family, and the agnostic setting where the manipulation graphs are chosen adversarially and not consistently modeled by a single graph in the family.

## 1 Introduction

Strategic considerations in machine learning have gained significant attention during the past decades. When ML algorithms are used to assist decisions that affect a strategic entity (e.g., a person, a company, or an LLM agent), this entity — henceforth the *agent* — naturally attempts to game the ML algorithms into making decisions that better serve the agent's goals, which in many cases differ from the decision maker's. Examples include loan applicants optimizing their credit score without actually improving their financial situation.[2] It is therefore desirable, if not imperative, that the ML algorithms used for decision-making be robust against such strategic manipulation.

Indeed, extensive effort has been made towards designing ML algorithms in the presence of strategic behavior, shaping the research area of *strategic machine learning* [Brückner and Scheffer, 2011, Hardt et al., 2016]. In particular, powerful frameworks have been proposed for *offline* environments, where the decision maker has access to historical data, on which they train a model that is subsequently used to make decisions about (i.e., to *classify*) members of a static population. These frameworks provide almost optimal learnability results and sample complexity bounds for strategic machine learning in offline environments, which gracefully generalize their non-strategic counterparts (see, e.g., [Zhang and Conitzer, 2021, Sundaram et al., 2023]).

---

*Authors are ordered alphabetically.

[2]The following article (among others) discusses some well-known tricks for this: `https://www.nerdwallet.com/article/finance/raise-credit-score-fast`.

38th Conference on Neural Information Processing Systems (NeurIPS 2024).

However, the situation becomes subtler in *online* environments, where the decision maker has little or no prior knowledge about the population being classified, and must constantly adjust the decision-making policy (i.e., the *classifier*) through trial and error. This is particularly challenging in the presence of strategic behavior, because often the decision maker can only observe the agent's features *after manipulation*. In such online environments, the performance of a learning algorithm is often measured by its *regret*, i.e., how many more mistakes it makes compared to the best classifier within a certain family *in hindsight*. For online strategic classification, while progress has been made in understanding the optimal regret in several important special cases, a full instance-wise characterization has been missing, even in the seemingly basic realizable setting (meaning that there always exists a perfect classifier in hindsight). This salient gap is the starting point of our investigation in this paper — which turns out to reach quite a bit beyond the gap itself.

Following prior work [Ahmadi et al., 2023, Cohen et al., 2024], we study the following standard and general model of online strategic classification: we have a (possibly infinite) feature space, equipped with a manipulation graph defined over it. An edge between two feature vectors $x_1$ and $x_2$ means that an agent whose true features are $x_1$ can pretend to have features $x_2$, and in fact, the agent would have incentives to do so if the label assigned to $x_2$ by the classifier is better than that assigned to the true features $x_1$. At each time step, the decision maker commits to a classifier (which may depend on observations from previous interactions), and an agent arrives and observes the classifier. The agent then responds to the classifier by reporting (possibly nontruthfully) a feature vector that leads to the most desirable label subject to the manipulation graph, i.e., the reported feature vector must be a neighbor of the agent's true feature vector. The decision maker then observes the reported feature vector, as well as whether the label assigned to that feature vector matches the agent's true label.

## 1.1   Our Results and Techniques

**An instance-optimal regret bound through the strategic Littlestone dimension.**   Our first main finding is an instance-optimal regret bound for online strategic classification in the realizable setting, when randomization is not allowed (we will discuss the role of randomization in Section 6). In this setting, there is a predefined hypothesis class of classifiers, in which there must exist one classifier that assigns all agents their true labels under manipulation. The decision maker, knowing that a perfect classifier exists in this class, tries to learn it on the fly while making as few mistakes as possible in the process. Naturally, the richer this hypothesis class is, the harder the decision maker's task will be (e.g., one extreme is when the hypothesis class contains only one classifier, and the decision maker knows a priori that that classifier must be perfectly correct, and the optimal regret is 0). Thus, the optimal regret must depend on the richness of the hypothesis class. Similarly, one can imagine that the optimal regret must also depend on the manipulation graph.

Previous work [Ahmadi et al., 2023, Cohen et al., 2024] has established regret bounds for this setting based on various complexity measures of the hypothesis class and the manipulation graph, including the size and the (classical) Littlestone dimension [Littlestone, 1988] of the hypothesis class, as well as the maximum out-degree of the manipulation graph. However, the optimality (when applicable) of these bounds only holds under the assumption that the mistake bound must be parametrized as a function on the classical Littlestone dimension and/or the graph's out-degree. However, these bounds are not tight for all instances, as there exist problem instances that are learnable where all these parameters are infinite.

To address the above issue, we introduce a new combinatorial complexity measure that generalizes the classical Littlestone dimension into strategic settings. Conceptually, the new notion also builds on the idea of "shattered trees", which has proved extremely useful in classical settings. However, the asymmetry introduced by strategic behavior[3] demands a much more delicate construction of shattered trees (among other intriguing implications to be discussed in Section 3). We show that the generalized Littlestone dimension captures precisely the optimal regret of any deterministic learning algorithm given a particular hypothesis class and a manipulation graph, thereby providing a complete characterization of learnability in this setting. Being instance-optimal, our bound strengthens and unifies all previous bounds for online strategic classification in the realizable setting.

---

[3]E.g., unlike in classical (non-strategic) settings, a true positive does not carry the same information as a false negative in our setting.

**An improved regret bound for the agnostic case.** We then proceed to the agnostic setting, where no hypothesis necessarily assigns correct labels to all agents. The regret is defined with respect to the best hypothesis in hindsight. Compared to the classical (i.e., non-strategic) setting, the main challenge is incomplete information: since the learner cannot observe original features, upon observing the behavior of an agent under one classifier, it is not always possible to counterfactually infer what would have been observed if the learner used another classifier.

To understand why this can be a major obstacle, recall some high-level ideas behind the algorithms for classical agnostic online classification (see, e.g., [Ben-David et al., 2009]). The key is to construct a finite set of "representative" experts out of the potentially infinite set of hypotheses, such that the best expert performs almost as well as the best hypothesis in hindsight. An agnostic learner then runs a no-regret learning algorithm (such as multiplicative weights) on the expert set, which in the long run matches the performance of the best expert, and in turn of the best hypothesis.

The partial information challenge appears in both steps of the above approach. First, to construct the set of representative experts, the learner needs to simulate the observation received by each hypothesis, had that hypothesis been used to label the strategic agents. Second, the no-regret algorithm on the expert set also needs to counterfacurally infer the agent's response to each expert. To circumvent both issues, we design a nuanced construction of the representative set of experts that effectively "guesses" each potential direction of the agent's manipulation. We then run the biased voting approach introduced in [Ahmadi et al., 2023] on the finite set of experts, which enjoys regret guarantees in the strategic setting even with partial information. Combined with our regret bound for the realizable setting, this approach yields an improved bound for the agnostic setting.

**Learning with unknown manipulation graphs.** Our last result focuses on relaxing the assumption that the learner has perfect knowledge about the manipulation graph structure. Instead, following previous works [Lechner et al., 2023, Cohen et al., 2024], we model their knowledge about the manipulation graph using a pre-defined graph class, which to some degree reflects the true set of feasible manipulations. In this setting, our work is the first that provides positive results when the learner only observes features after manipulation. We start with the realizable setting where the manipulation graphs are consistently modeled by the same (unknown) graph in the class. In this setting, we provide a more careful construction of the representative experts to account for the additional challenge of unknown graphs. Combing this construction with re-examining the effectiveness of the biased voting approach [Ahmadi et al., 2023] when the input graph is a overly-conservative estimate of the true graph, we obtain the first regret bound in this setting that is approximately optimal (up to logarithmic factors) in certain instances. We also extend our results to fully agnostic settings where the agents in each round manipulates according to a potentially different graph, and the best graph in the class has nonzero error in modeling all the manipulations.

## 1.2 Further Related Work

There is a growing line of research that studies learning from data provided by strategic agents [Dalvi et al., 2004, Dekel et al., 2008, Brückner and Scheffer, 2011]. The seminal work of Hardt et al. [2016] introduced the problem of *strategic classification* as a repeated game between a mechanism designer that deploys a classifier and an agent that best responds to the classifier by modifying their features at a cost. Follow-up work studied different variations of this model, in an online learning setting [Dong et al., 2018, Chen et al., 2020, Ahmadi et al., 2021], incentivizing agents to take improvement actions rather than gaming actions [Kleinberg and Raghavan, 2020, Haghtalab et al., 2020, Alon et al., 2020, Ahmadi et al., 2022], causal learning [Bechavod et al., 2021, Perdomo et al., 2020], screening processes [Cohen et al., 2023], fairness [Hu et al., 2019], etc.

Two different models for capturing the set of plausible manipulations have been considered in the literature. The first one is a geometric model, where the agent's best-response to the mechanism designer's deployed classifier is a state within a bounded distance (with respect to some $\ell_p$ norm) from the original state, i.e. feature set [Dong et al., 2018, Chen et al., 2020, Shao et al., 2024, Sundaram et al., 2023, Ghalme et al., 2021, Haghtalab et al., 2020]. In the second model, introduced by Zhang and Conitzer [2021] there is a combinatorial structure, i.e. manipulation graph, that captures the agent's set of plausible manipulations. This model has been studied in both offline PAC learning [Lechner and Urner, 2022, Zhang and Conitzer, 2021, Lechner et al., 2023] and online settings [Ahmadi et al., 2023]. In a recent work, Lechner et al. [2023] consider this problem in an

offline setting where the underlying manipulation is unknown and belongs to a known family of graphs. Our work improves the results given by [Ahmadi et al., 2023] in the online setting and also extends their results to the setting where the underlying manipulation is unknown and belongs to a known family of graphs.

Our work is also closely related to that of Cohen et al. [2024], with two main points of distinction. First, in the realizable setting, their bound is shown to be optimal for a specific instance, whereas our bound is instance-wise optimal. Second, in both the agnostic and the unknown graph settings, they assume that agents' original features are observable before the learner makes decisions, whereas our algorithm only requires access to post-manipulation features.

Finally, our work is also tangentially connected to several recent advances in understanding multi-class classification under bandit feedback, e.g., [Raman et al., 2024, Filmus et al., 2024], as the false-positive mistake types can be treated as multiple labels at a very abstract level. However, an additional challenge in the strategic setting is that the learner needs to choose a classifier without observing the original instance to be labeled.

# 2 Model and Preliminaries

## 2.1 Strategic classification.

Let $\mathcal{X}$ be a space of feature vectors, $\mathcal{Y} = \{-1, 1\}$ be the binary label space, and $\mathcal{H} : \mathcal{X} \to \mathcal{Y}$ be a hypothesis class that is known to the learner (also referred to as "decision-makers"). In the strategic classification setting, agents prefer positive labels over negative labels, and they may manipulate their features within a predefined range to receive a positive label. We use the *manipulation graphs* introduced by [Zhang and Conitzer, 2021, Lechner and Urner, 2022] to model the set of feasible manipulations. The manipulation graph $G(\mathcal{X}, \mathcal{E})$ is a directed graph in which each node corresponds to a feature vector in $\mathcal{X}$, and each edge $(x_1, x_2) \in \mathcal{E} \subseteq \mathcal{X}^2$ represents that an agent with initial feature vector $x_1$ can modify their feature vector to $x_2$. For each $x \in \mathcal{X}$, we use $N_G^+(x)$ to denote the set of out-neighbors of $x$ in $G$, excluding $x$ itself, and $N_G^+[x]$ to denote the out-neighbors including $x$. Formally, $N_G^+(x) = \{x' \in \mathcal{X} \setminus \{x\} \mid (x, x') \in \mathcal{E}\}$ and $N_G^+[x] = \{x\} \cup N_G^+(x)$. Similarly, we use $N_G^-(x)$ and $N_G^-[x]$ to denote respectively the exclusive and inclusive in-neighborhood of $x$.

**Agents' utility and the manipulation rule.** Given a manipulation graph $G(\mathcal{X}, \mathcal{E})$ and a classifier $h \in \mathcal{Y}^{\mathcal{X}}$, an agent with initial features $x \in \mathcal{X}$ aims to maximize their utility by potentially manipulating their features to a different $x'$. The agent's utility function $\mathsf{util}_{G,h}(x, x')$ is defined as

$$\mathsf{util}_{G,h}(x, x') = h(x') - \infty \cdot \mathbb{1}\{(x, x') \in \mathcal{E}\},$$

where the agent's utility is the classification outcome $h(x')$ minus the manipulation cost associated with changing features from $x$ to $x'$. For the classification outcome, agents receive utility $+1$ if the classifier $h$ labels $x'$ as positive and $-1$ otherwise. For the manipulation cost, moving from $x$ to $x'$ incurs no cost if the two features are connected by an edge in $G$, but when $x$ and $x'$ are not connected (i.e., $(x, x') \notin \mathcal{E}$), the manipulation incurs an infinite cost, effectively making such a manipulation infeasible. As a result, an agent with initial features $x$ would move to some $x'$ in the inclusive neighborhood $N_G^+[x]$ that is labeled as positive by $h$, if such a node exists.

Formally, the set of best response features from $x$, denoted by $\mathsf{BR}_{G,h}(x)$, is defined as

$$\mathsf{BR}_{G,h}(x) \triangleq \{x' \mid \mathsf{util}_{G,h}(x, x') = +1\} = N_G^+[x] \cap \{x \mid h(x) = +1\}.$$

If, however, the set $\mathsf{BR}_{G,h}(x)$ is empty — indicating that the entire out-neighborhood $N_G^+[x]$ is labeled as negative by $h$ — then the agent is assumed to *not manipulate* their features and remain at $x$. In addition, when there are multiple nodes in $\mathsf{BR}_{G,h}(x)$, the agent may select any node arbitrarily. We do not require the selection to be consistent across rounds or to follow a pre-specified rule. Our results specifically focus on the learner's mistakes against worst-case (adversarial) selections by the agent.[4] Finally, the manipulated feature vector is denoted by $\mathsf{br}_{G,h}(x)$.

---

[4] Some previous works, such as [Ahmadi et al., 2023, Cohen et al., 2024], assume that agents will not manipulate their features if $x \in \mathsf{BR}_{G,h}(x)$. While this assumption enforces consistency in the special case of $h(x) = 1$, we remark that most algorithms proposed by both papers remain effective even without this assumption.

The labels induced by the manipulated feature vectors $\mathsf{br}_{G,h}(x)$ are captured by *effective classifiers* $\widetilde{h}_G$, formally defined as

$$\widetilde{h}_G(x) \triangleq h(\mathsf{br}_{G,h}(x)) = \begin{cases} +1, & \text{if } \exists v \in N_G^+[x], \text{ s.t. } h(v) = +1; \\ -1, & \text{otherwise.} \end{cases}$$

**Online learning.** We consider an *online* strategic classification setting modeled as a repeated game between the learner (aka the decision maker) and an adversary over $T$ rounds, where the learner make decisions according to an online learning algorithm $\mathcal{A}$. At each round $t \in [T]$, the learner first commits to the classifier $h_t \in \mathcal{Y}^{\mathcal{X}}$ (not necessarily restricted to $\mathcal{H}$) that is generated by $\mathcal{A}$. The adversary then selects an agent $(x_t, y_t)$ where $x_t \in \mathcal{X}$ is the original feature vector and $y_t \in \mathcal{Y}$ is the true label. In response to $h_t$, the agent manipulates their features from $x_t$ to $v_t = \mathsf{br}_{G,h}(x_t)$. Consequently, the learner observes the manipulated features $v_t$ (instead of $x_t$), and incurs a mistake if $y_t \neq h_t(v_t)$. We use $S = (x_t, y_t)_{t \in [T]}$ to denote the sequence of agents.

The learner aims to minimize the Stackelberg regret on $S$ with respect to the optimal hypothesis $h^\star \in \mathcal{H}$ had the agents responded to $h^\star$:

$$\mathsf{Regret}_{\mathcal{A}}(S, \mathcal{H}, G) \triangleq \sum_{t=1}^T \mathbb{1}\{h_t(\mathsf{br}_{G,h_t}(x_t)) \neq y_t\} - \min_{h^\star \in \mathcal{H}} \sum_{t=1}^T \mathbb{1}\{h^\star(\mathsf{br}_{G,h^\star}(x_t)) \neq y_t\}.$$

We call a sequence $S$ *realizable* with respect to $\mathcal{H}$ if the optimal-in-hindsight hypothesis $h^\star \in \mathcal{H}$ achieves zero mistakes on $S$. Specifically, this means that for all $(x_t, y_t)$ in the sequence $S$, we have $y_t = h^\star(\mathsf{br}_{G,h^\star}(x_t)) = \widetilde{h}_G^\star(x_t)$. In such cases, the learner's regret coincides with the number of mistakes made. We use $\mathsf{Mistake}_{\mathcal{A}}(\mathcal{H}, G)$ to denote the *maximal number of mistakes* that $\mathcal{A}$ makes against any realizable sequence with respect to class $\mathcal{H}$ and graph $G$. A deterministic algorithm is called *minmax optimal* or *instance-optimal* if achieves the minimal $\mathsf{Mistake}_{\mathcal{A}}(\mathcal{H}, G)$ across all deterministic algorithms[5]. We denote this optimal mistake bound by $\mathcal{M}(\mathcal{H}, G) \triangleq \inf_{\mathcal{A} \text{ deterministic}} \mathsf{Mistake}_{\mathcal{A}}(\mathcal{H}, G)$.

## 2.2 Classical Littlestone Dimension

In this section, we revisit the classical online binary classification setting where the agents are unable to strategically manipulate their features. This setting can be viewed as a special case of strategic classification where the manipulation graph $G$ consists solely of isolated nodes. We will introduce the characterization of the optimal mistake in this classical setting — known as the *Littlestone Dimension* — which inspires our analysis in the strategic setting.

**Definition 2.1** ($\mathcal{H}$-Shattered Littlestone Tree). *A Littlestone tree shattered by hypothesis class $\mathcal{H}$ of depth $d$ is a binary tree where:*

- *(Structure) Nodes are labeled by $\mathcal{X}$ and each non-leaf node has exactly two outgoing edges that are labeled by $+1$ and $-1$, respectively.*

- *(Consistency) For every root-to-leaf path $x_1 \xrightarrow{y_1} x_2 \xrightarrow{y_2} \cdots x_d \xrightarrow{y_d} x_{d+1}$ where $x_1$ is the root node and each $y_t$ is the edge connecting $x_t$ and $x_{t+1}$, there exists a hypothesis $h \in \mathcal{H}$ that is consistent with the entire path, i.e., $\forall t \leq d$, $h(x_t) = y_t$.*

The above tree structure intuitively models an adversary's strategy to maximize the learner's mistakes, where each node $x_t$ represents the unlabeled instance to be presented to the learner, and $y_t$ represents the type of mistake (either a false positive or a false negative) that the adversary aims to induce. For example, if the learner predicts the label of $x_t$ to be $\widehat{y}_t = +1$, then the adversary will declare $y_t = -1$, enforce a false positive mistake, and choose the next instance $x_{t+1}$ as the children of the current node along the $-1$ edge. In addition, the *consistency* requirement guarantees that the resulting input sequence is realizable by some classifier in $\mathcal{H}$.

**Definition 2.2** (Littlestone Dimension). *The Littlestone dimension of class $\mathcal{H}$, denoted as $\mathsf{Ldim}(\mathcal{H})$, is the maximum integer $d$ such that there exists an $\mathcal{H}$-shattered Littlestone of depth $d$.*

The interpretation of the true structure immediately implies that the mistake of any algorithm should be lower bounded by $\mathsf{Ldim}(\mathcal{H})$. Moreover, a seminal result by Littlestone [1988] also showed that an online learning algorithm known as the Standard Optimal Algorithm (SOA, see Algorithm 2 in

---

[5]In this paper, we mainly consider deterministic algorithms. We will discuss the role of randomness in Section 6 and Appendix E.

Appendix A) can achieve this lower bound. Together, they form a complete characterization of the optimal mistake bound in the classical setting, which we summarize in the following proposition.

**Proposition 2.1** (Optimal Mistake Bound [Littlestone, 1988]). *Let $\mathcal{M}(\mathcal{H})$ be the optimal mistake in the classical online learning setting, then $\mathcal{M}(\mathcal{H}) = \mathsf{Ldim}(\mathcal{H})$.*

In the next section, we will discuss the challenges of extending Littlestone's characterization to the strategic setting, and present our solution.

## 3 The Strategic Littlestone Dimension

In this section, we will introduce a new combinatorial dimension called the *Strategic Littlestone Dimension*, and show that it characterizes the minmax optimal mistake bound for strategic classification.

Inspired by the classical Littlestone dimension, we hope to use a tree structure to model an adversary's strategy for selecting agents $(x_t, y_t)$, where nodes serve as (proxies of) the initial feature vector of each agent, and edges represent the types of mistakes that the adversary can induce. However, since agents can strategically manipulate their features, the potential mistakes associated with the same initial feature vector could manifest in many more types depending on the learner's choice of classifiers. Specifically, let $x$ be the initial feature vector. Then a mistake associated with $x$ might be observed as a false negative mistake at node $x$ (denoted as $(x, +1)$ where $x$ is the observable node and $+1$ is the true label), or as a false positive mistake at any outgoing neighbor $v$ of $x$ (denoted as $(v, -1)$ accordingly). Therefore, an adversary's strategy should accommodate all such possibilities, which necessitates the strategic Littlestone tree to contain branches representing all potential mistake types.

Another challenge is caused by the mismatch of the information available to the learner and the adversary. Since the learner only observes manipulated features instead of the true ones, the amounts of information carried by false positive and false negative mistakes are inherently asymmetric. False negatives provide full-information feedback as the manipulated and original features are identical. However, false positives introduce uncertainty about the original features, which could potentially be any in-neighbor of the observed one. As a result, a hypothesis is deemed "consistent" with a false positive observation as long as it can correctly label any one of the potential original nodes.

Now we formally introduce the *strategic Littlestone tree* with adapted branching and consistency rules tailored for strategic classification. See Figure 1 for a pictorial illustration.

**Definition 3.1** ($\mathcal{H}$-Shattered Strategic Littlestone Tree). *A Strategic Littlestone tree for hypothesis class $\mathcal{H}$ under graph $G$ with depth $d$ is a tree where:*

- *(Structure) Nodes are labeled by $\mathcal{X}$. The set of outgoing edges from each non-leaf node $x$ are: one false negative edge $(x, +1)$, and a set of false positive edges $\{(v, -1) \mid v \in N_G^+[x]\}$.*

- *(Consistency) For every root-to-leaf path $x_1' \xrightarrow{(v_1, y_1)} x_2' \xrightarrow{(v_2, y_2)} \cdots x_d' \xrightarrow{(v_d, y_d)} x_{d+1}'$ where $x_1'$ is the root node and $(v_t, y_t) \in \mathcal{X} \times \{\pm 1\}$ is the edge that connects $x_t'$ and $x_{t+1}'$, there exists a hypothesis $h \in \mathcal{H}$ that is consistent with the entire path. Specifically, $\forall t \leq d, \exists x_t$ s.t. $\widetilde{h}_G(x_t) = y_t$, where $x_t$ satisfies $x_t = v_t$ if $y_t = +1$, and $x_t \in N_G^{-1}[v_t]$ if $y_t = -1$.*

**Definition 3.2** (Strategic Littlestone Dimension). *The Strategic Littlestone Dimension of a hypothesis class $\mathcal{H}$ under graph $G$, denoted with $\mathsf{SLdim}(\mathcal{H}, G)$, is defined as the largest nonnegative integer $d$ for which there exists a Strategic Littlestone tree of depth $d$ shattered by $\mathcal{H}$ under graph $G$.*

**Theorem 3.1** (Minmax optimal mistake for strategic classification). *For any hypotheses class $\mathcal{H}$ and manipulation graph $G$, the minmax optimal mistake in the realizable setting is captured by the strategic Littlestone dimension, i.e., $\mathcal{M}(\mathcal{H}, G) = \mathsf{SLdim}(\mathcal{H}, G)$.*

We divide the proof of Theorem 3.1 into two parts: the lower bound direction is established in Theorem 3.2, and the upper bound direction is established in Theorem 3.3.

**Theorem 3.2** (Lower bound part of Theorem 3.1). *For any pair of hypothesis class $\mathcal{H}$ and manipulation graph $G$, any deterministic online learning algorithm $\mathcal{A}$ must suffer a mistake lower bound of $\mathsf{Mistake}_\mathcal{A}(\mathcal{H}, G) \geq \mathsf{SLdim}(\mathcal{H}, G)$.*

*Proof sketch of Theorem 3.2.* Let $\mathcal{T}$ be a strategic Littlestone tree for $(\mathcal{H}, G)$ with depth $d$. We will show that for any deterministic algorithm $\mathcal{A}$, there exists an adversarial sequence of agents

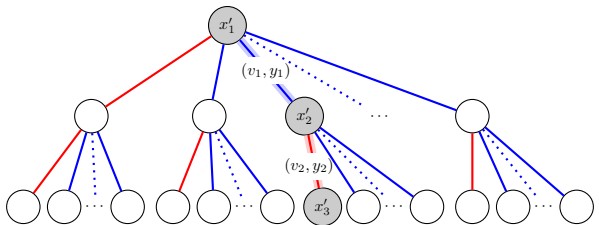

Figure 1: A Strategic Littlestone Tree with depth 2. False negative edges are marked red, whereas false positive edges are marked blue. The highlighted path $x_1' \xrightarrow{(v_1,y_1)} x_2' \xrightarrow{(v_2,y_2)} x_3'$ is an example root-to-leaf path. In this path, the first observation $(v_1, y_1)$ is a false positive, which satisfies $v_1 \in N_G^+[x_1']$ and $y_1 = -1$; the second observation $(v_2, y_2)$ is a false negative, which satisfies $v_2 = x_2'$ and $y_2 = +1$.

$S = (x_t, y_t)_{t \in [d]}$ such that $\mathcal{A}$ is forced to make a mistake at every round. We construct the sequence $S$ by first finding a path $x_1' \xrightarrow{(v_1,y_1)} x_2' \xrightarrow{(v_2,y_2)} \cdots x_d' \xrightarrow{(v_d,y_d)} x_{d+1}'$ in tree $\mathcal{T}$ which specifies the types of mistakes that the adversary wishes to induce, then reverse-engineering this path to obtain the sequence of initial feature vectors before manipulation that is realizable under $\mathcal{H}$. We remark that the need for reverse-engineering is unique to the strategic setting, which is essential in resolving the information asymmetry between the learner and the adversary regarding the true features. We formally prove this theorem in Appendix B.1. □

In the remainder of this section, we present an algorithm called the Strategic Standard Optimal Algorithm (SSOA) that achieves the instance-optimal mistake bound of $\mathsf{SLdim}(\mathcal{H}, G)$. We first define some notations. For any hypothesis sub-class $\mathcal{F} \subset \mathcal{H}$ and an observable labeled instance $(v, y) \in \mathcal{X} \times \mathcal{Y}$, we use $\mathcal{F}_G^{(v,y)}$ to denote the subset of $\mathcal{F}$ that is consistent with $(v, y)$ under manipulation graph $G$. We refer to the consistency rule defined in Definition 3.1, i.e., for all $v \in \mathcal{X}$, $\mathcal{F}_G^{(v,+1)}$ and $\mathcal{F}_G^{(v,-1)}$ are defined respectively as:

$$\mathcal{F}_G^{(v,+1)} \triangleq \{h \in \mathcal{F} \mid \widetilde{h}_G(v) = +1\}; \qquad \mathcal{F}_G^{(v,-1)} \triangleq \{h \in \mathcal{F} \mid \exists x \in N_G^-[v] \text{ s.t. } \widetilde{h}_G(x) = -1\}.$$

We present SSOA in Algorithm 1 and prove its optimality in Theorem 3.3. The high-level idea of SSOA is similar to the classical SOA algorithm (Algorithm 2): it maintains a version space of classifiers consistent with the history, and chooses a classifier $h_t$ in a way that guarantees the "progress on mistakes" property. This means that the (strategic) Littlestone dimension of the version space should decrease whenever a mistake is made.

However, designing $h_t$ to satisfy this property in a strategic setting is more challenging because the potential types of mistakes depend on $h_t$'s labeling in the neighborhood $N_G^+[x_t]$, where $x_t$ is unobservable. If we directly optimize the labelings on $N_G^+[x]$ for each $x$ independently, the resulting classifier may suggest self-contradictory labelings to the nodes in the overlapping parts of $N_G^+[x]$ and $N_G^+[x']$ for different $x$ and $x'$. Instead, the learner needs to choose a single $h_t$ that simultaneously guarantees the "progress on mistakes" property for all possible $x_t$ and their neighborhoods.

In the following, we will show that this challenge can be resolved by choosing a classifier $h_t$ that labels each node $x$ only based on whether a false positive observation $(x, -1)$ can decrease the strategic Littlestone dimension, as described in Line 3 of Algorithm 1.

**Theorem 3.3** (Upper bound part of Theorem 3.1). *The* SSOA *algorithm (Algorithm 1) achieves a maximal mistake bound of* $\mathsf{Mistake}_{\mathsf{SSOA}}(\mathcal{H}, G) \leq \mathsf{SLdim}(\mathcal{H}, G)$.

**Remark 3.4** (Comparison with previous results). *Since the mistake bound of* SSOA *is shown to be instance-optimal across all deterministic algorithms, it improves upon the bounds established by Ahmadi et al. [2023], Cohen et al. [2024], which both depend on the maximum out-degree of the graph $G$. Furthermore, we show in Appendix B.2 that the gap between their bounds and ours could be arbitrarily large. An extreme example is the complete graph $G$ supported on an unbounded domain, where* $\mathsf{SLdim}(\mathcal{H}, G) = 1$ *but both previous bounds are $\infty$.*

*Proof of Theorem 3.3.* It suffices to prove that if SSOA makes a mistake at round $t$, then the strategic Littlestone dimension of version space $\mathcal{H}_t$ (maintained by the SSOA algorithm in Line 5) must

---

**Algorithm 1:** The Strategic Standard Optimal Algorithm (SSOA)

---

**Input:** Hypothesis class $\mathcal{H}$, manipulation graph $G$.
**Initialization:** Version space $\mathcal{H}_0 \leftarrow \mathcal{H}$.

1 **for** $t \in [T]$ **do**
2     Commit to the classifier $h_t : \mathcal{X} \rightarrow \{\pm 1\}$ defined as follows:
3     $\forall x \in \mathcal{X}, h_t(x) \leftarrow \begin{cases} +1, & \text{if } \mathsf{SLdim}\left((\mathcal{H}_{t-1})_G^{(x,-1)}, G\right) < \mathsf{SLdim}(\mathcal{H}_{t-1}, G); \\ -1, & \text{otherwise.} \end{cases}$ ;
4     Observe the manipulated feature vector $v_t$ and the true label $y_t$;
5     If a mistake occurs $(h_t(v_t) \neq y_t)$, update $\mathcal{H}_t \leftarrow (\mathcal{H}_{t-1})_G^{(v_t, y_t)}$. Otherwise $\mathcal{H}_t \leftarrow \mathcal{H}_{t-1}$.
6 **end**

---

decrease by at least 1, namely $\mathsf{SLdim}(\mathcal{H}_t, G) \leq \mathsf{SLdim}(\mathcal{H}_{t-1}, G) - 1$. For notational convenience, let $d = \mathsf{SLdim}(\mathcal{H}_{t-1}, G)$.

**False positives.** We start with the case where SSOA makes a false positive mistake, i.e., $h_t(v_t) = +1$ but $y_t = -1$. According to the definition of classifier $h_t$ and the update rule of version space $\mathcal{H}_t$, we immediately obtain

$$\mathsf{SLdim}(\mathcal{H}_t, G) = \mathsf{SLdim}\left((\mathcal{H}_{t-1})_G^{(v_t, -1)}, G\right) < \mathsf{SLdim}(\mathcal{H}_{t-1}, G) \;\Rightarrow\; \mathsf{SLdim}(\mathcal{H}_t, G) \leq d - 1.$$

**False negatives.** Then we consider the case where SSOA makes a false negative mistake, i.e., $h_t(v_t) = -1$ but $y_t = +1$. For the sake of contradiction, assume that the strategic Littlestone dimension does not decrease, i.e., $\mathsf{SLdim}(\mathcal{H}_t, G) = \mathsf{SLdim}\left((\mathcal{H}_{t-1})_G^{(v_t, +1)}, G\right) = d$. This assumption implies that there exists a strategic Littlestone tree $\mathcal{T}$ that is shattered by $(\mathcal{H}_{t-1})_G^{(v_t, +1)}$ and of depth $\mathsf{SLdim}(\mathcal{H}_{t-1}, G)$.

Since the agent is classified as negative, it must be the case that the agent has not manipulated (i.e., $x_t = v_t$), and the entire outgoing neighborhood $N_G^+[x_t]$ is labeled as negative by $h_t$. Therefore, according to the definition of $h_t$, for all $v \in N_G^+[x_t]$, we have $\mathsf{SLdim}\left((\mathcal{H}_{t-1})_G^{(v,-1)}, G\right) = d$, which implies that there also exists a strategic Littlestone tree $\mathcal{T}_v$ of depth $d$ that is shattered by $(\mathcal{H}_{t-1})_G^{(v,-1)}$.

Now consider the tree $\mathcal{T}'$ with root $x_t$, subtree $\mathcal{T}$ on the false negative edge $(x_t, +1)$, and subtree $\mathcal{T}_v$ on each false positive edge $(v, -1)$ for all $v \in N_G^+[x_t]$. Since we have argued that each subtree has depth $d$, the overall depth of $\mathcal{T}'$ is $d + 1$. We claim that $\mathcal{T}'$ is shattered by $\mathcal{H}_{t-1}$. In fact, for all root-to-leaf paths in $\mathcal{T}'$, the first observation is guaranteed to be consistent with all hypotheses in the subclass for the subtree, and the consistency of each subtree ensures the existence of a hypothesis that makes all subsequent observations realizable.

We have thus constructed a strategic Littlestone tree $\mathcal{T}'$ that is shattered by $\mathcal{H}_{t-1}$ and of depth $d + 1$. However, this contradicts with the assumption that $\mathsf{SLdim}(\mathcal{H}_{t-1}, G) = d$. Therefore, it must follow that $\mathsf{SLdim}(\mathcal{H}_t, G) \leq d - 1 = \mathsf{SLdim}(\mathcal{H}_{t-1}, G) - 1$, which in turn proves $\mathsf{Mistake}_{\mathsf{SSOA}}(\mathcal{H}, G) \leq \mathsf{SLdim}(\mathcal{H}, G)$. $\square$

## 4 Agnostic Setting

In this section, we study the regret bound in the agnostic setting. Recall that benchmark is defined as the minimum number of mistakes that the best hypothesis in $\mathcal{H}$ makes, i.e., $\mathsf{OPT} \triangleq \min_{h^\star \in \mathcal{H}} \sum_{t \in [T]} \mathbb{1}\{h^\star(\mathsf{br}_{G, h^\star}(x_t)) \neq y_t\}$. We will present an algorithm that has vanishing regret compared to $\Delta_G^+ \cdot \mathsf{OPT}$ whenever the strategic Littlestone dimension is bounded, where $\Delta_G^+$ is the maximum out-degree of $G$. Inspired by the classical reduction framework proposed by Ben-David et al. [2009], our algorithm aims to reduce the agnostic problem to that of strategic online learning with expert advice by constructing a finite number of representative experts that performs almost as well as the potentially unbounded hypothesis class. The problem with a finite expert set can then

be solved using the biased weighted voting algorithm proposed by Ahmadi et al. [2023]. However, establishing the reduction turns out to be more challenging in the strategic setting, as the learner can only observe manipulated features instead of the original ones. We address this problem by designing the experts to "guess" every possibile direction the original node could have come. We present our algorithms (Algorithms 3 and 4) in Appendix C and analyze their regret in Theorem 4.1.

**Theorem 4.1.** *For any adaptive adversarial sequence $S$ of length $T$, the Agnostic Online Strategic Classification algorithm (Algorithm 3) has regret bound*

$$\mathsf{Regret}(S, \mathcal{H}, G) \leq O\left(\Delta_G^+ \cdot \mathsf{OPT} + \Delta_G^+ \cdot \mathsf{SLdim}(\mathcal{H}, G) \cdot (\log T + \log \Delta_G^-)\right),$$

*where $\Delta_G^+$ (resp. $\Delta_G^-$) denotes the maximum out-degree (resp. in-degree) of graph $G$.*

**Remark 4.2.** *Ahmadi et al. [2023] showed that there exists instances in which all deterministic algorithms must suffer regret $\Omega(\Delta_G^+ \cdot \mathsf{OPT})$, which means the first term in the above bound is necessary. The second term connects to our instance-wise lower bound of $\mathsf{SLdim}(\mathcal{H}, G)$ in Theorem 3.2.*

*Proof sketch of Theorem 4.1.* We use $\mathfrak{E}$ to denote the set of experts constructed in Algorithm 4, and define $\mathsf{OPT}^{\mathfrak{E}}$ as the minimum number of mistakes made by the best expert $\mathfrak{e}^\star \in \mathfrak{E}$, had the agents responded to $\mathfrak{e}^\star$. Then the *Biased Weighted Majority Vote* algorithm from Ahmadi et al. [2023] guarantees that the number of mistakes made by Algorithm 3 is at most $\Delta_G^+ \cdot \mathsf{OPT}^{\mathfrak{E}} + \Delta_G^+ \cdot \log |\mathfrak{E}|$. According to our construction of experts, the total number of experts satisfies $\log |\mathfrak{E}| \leq \log\left(\sum_{m \leq d} \binom{T}{m} \cdot (\Delta_G^-)^m\right) \leq O(d \cdot (\log T + \log \Delta_G^-))$, where $d = \mathsf{SLdim}(\mathcal{H}, G)$ is the strategic Littlestone dimension. Therefore, it suffices to show that $\mathsf{OPT}^{\mathfrak{E}}$ is not too much larger than $\mathsf{OPT}$—in other words, the set of experts $\mathfrak{E}$ are *representative* enough of the original hypothesis class $\mathcal{H}$ in their ability of performing strategic classification. We use the following lemma, which we prove in Appendix C by establishing the equivalence between the SSOA instance running on the sequence labeled by the effective classifier and the SSOA instance simulated by a specific expert.

**Lemma 4.3** (Experts are representative). *For any hypothesis $h \in \mathcal{H}$ and any sequence of agents $S$, there exists an expert $\mathfrak{e}_h \in \mathfrak{E}$ that makes at most $\mathsf{SLdim}(\mathcal{H}, G)$ more mistakes than $h$.*

## 5 Unknown Manipulation Graph

In this section, we generalize the main settings to relax the assumption that the learner has full knowledge about the underlying manipulation graph $G$. Instead, we use a graph class $\mathcal{G}$ to capture the learner's knowledge about the manipulation graph. In Section 5.1, we begin with the *realizable graph class* setting, where the true manipulation graph remains the same across rounds and belongs to the family $\mathcal{G}$. We then study the *agnostic graph class* setting in Section 5.2, where we drop both assumptions and allow our regret bound to depend on the "imperfectness" of $\mathcal{G}$. In both cases, we assume the hypothesis class $\mathcal{H}$ is also agnostic, which encompasses the setting where $\mathcal{H}$ is realizable.

### 5.1 Realizable graph classes

In this section, we assume that there exists a perfect (but unknown) graph $G^\star \in \mathcal{G}$, such that each agent $(x_t, y_t) \in S$ manipulates according to $G^\star$. We define the benchmark $\mathsf{OPT}_{\mathcal{H}}$ to be the optimal number of mistakes made by the best $h^\star \in \mathcal{H}$ assuming that each agent best responds to $h^\star$ according to $G^\star$. Formally, $\mathsf{OPT}_{\mathcal{H}} \triangleq \min_{h^\star \in \mathcal{H}} \mathbb{1}\{h^\star(\mathsf{br}_{G^\star, h^\star}(x_t)) \neq y_t\}$. Same to our main setting, we assume that the learner only observes the post-manipulation features $v_t = \mathsf{br}_{G^\star, h_t}(x_t)$ after they commit to classifier $h_t$, but cannot observe the original features $x_t$.

Our algorithm (Algorithm 6) for this setting leverages two main ideas. First, to overcome the challenge that $G^\star$ is unknown to the experts, we blow up the number of experts by a factor of $|\mathcal{G}|$ and let each expert simulate their own SSOA instance according to some internal belief of $G^\star$. Since the regret bound depends logarithmic on the number of experts, this only introduces an extra $\log |\mathcal{G}|$ term, which has been shown by Cohen et al. [2024] to be unavoidable even when the learner has access to the original features.

Our second idea involves re-examining the correctness of Algorithm 5 for bounded expert class to the scenario where the input $G$ is a pessimistic estimate of the true graph $G^\star$, i.e., $G$ contains all the edges in $G^\star$ but potentially some extra edges. This allows us to use $G_{\mathrm{union}}$ whose edge set is taken to

be the union of all egdes in $\mathcal{G}$. Combining these two ideas, we present our algorithm and establish its regret bound (Theorem 5.1) in Appendix D.

**Theorem 5.1.** *For any realizable graph class $\mathcal{G}$ and any adaptive adversarial sequence $S$ of length $T$, Algorithm 6 has regret bound*

$$\text{Regret}(S, \mathcal{H}, G) \leq O\left(\Delta_{\mathcal{G}}^+ \cdot \left(\text{OPT}_{\mathcal{H}} + d_{\mathcal{G}} \cdot (\log T + \log \Delta_{\mathcal{G}}^-) + \log |\mathcal{G}|\right)\right),$$

*where $d_{\mathcal{G}} \triangleq \max_{G \in \mathcal{G}} \text{SLdim}(\mathcal{H}, G)$ is the maximum strategic Littlestone dimension for all graphs in $\mathcal{G}$, $\Delta_{\mathcal{G}}^+ \triangleq \Delta_{G_{union}}^+$ is the maximum out-degree of $G_{union}$ (i.e., the union of graphs in $\mathcal{G}$), and $\Delta_{\mathcal{G}}^- \triangleq \max_{G \in \mathcal{G}} \Delta_G^-$ is the maximum max in-degree over graphs in $G$.*

**Remark 5.2** (Implications in the realizable setting)**.** *In the realizable setting where $\text{OPT}_{\mathcal{H}} = 0$, Theorem 5.1 implies a mistake bound of $\tilde{O}(\Delta_{\mathcal{G}}^+ \cdot d_{\mathcal{G}} + \log |\mathcal{G}|)$. This bound is optimal up to logarithmic factors due to a lower bound proved by Cohen et al. [2024, Proposition 14]. They constructed an instance with $|\mathcal{G}| = |\mathcal{H}| = \Theta(n)$ in which any deterministic algorithm makes $\Omega(n)$ mistakes. In this instance, our bound evaluates to be $\tilde{O}(n)$ since $\Delta_{\mathcal{G}}^+ = \Theta(n)$ and $d_{\mathcal{G}} = 1$.*

### 5.2 Agnostic graph classes

In this section, we consider a fully agnostic setting where each agent $(x_t, y_t)$ may behave according to a different manipulation graph $G_t \subseteq G_{union}$. We define the benchmark $\text{OPT}_{\mathcal{G}}$ to count the number of times that the best graph $G^\star \in \mathcal{G}$ fails to model the local manipulation structure under $G_t$, and $\text{OPT}_{\mathcal{H}}$ is defined as in Section 5.1, using the graph $G^\star$ that achieves $\text{OPT}_{\mathcal{G}}$.

$$\text{OPT}_{\mathcal{G}} \triangleq \min_{G^\star \in \mathcal{G}} \sum_{t=1}^T \mathbb{1}\left\{N_{G^\star}^+[x_t] \neq N_{G_t}^+[x_t]\right\}, \quad \text{OPT}_{\mathcal{H}} \triangleq \min_{h^\star \in \mathcal{H}} \sum_{t=1}^T \mathbb{1}\{h^\star(\text{br}_{G^\star, h^\star}(x_t) \neq y_t)\}\,^6.$$

Assuming access to an upper bound $N$ of $\text{OPT}_{\mathcal{G}}$, we present Algorithm 7 that achieves a regret bound of $\tilde{O}\left(\Delta_{\mathcal{G}}^+(N + \text{OPT}_{\mathcal{H}} + d_{\mathcal{G}})\right)$, as shown in Theorem D.2. We additionally apply the standard doubling trick to remove the requirement of knowing $N$. More details can be found in Appendix D.2.

## 6 Discussion and Future Research

**Improved bounds for the agnostic setting.** An immediate direction for future research is tightening our bounds in the agnostic setting under known manipulation graph. Note that our upper bound is $\tilde{O}\left(\Delta_G^+ \cdot (\text{OPT} + \text{SLdim}(\mathcal{H}, G))\right)$ whereas the lower bounds are $\Omega(\Delta_G^+ \cdot \text{OPT})$ from Ahmadi et al. [2023] and $\Omega(\text{SLdim}(\mathcal{H}, G))$ from Theorem 3.2. The extra $\Delta_G^+$ factor is introduced by the strategic learning-with-expert-advice algorithm, for which all known results have the dependency on $\Delta_G^+$.

**Randomized learners.** Our results mainly focus on deterministic learners. It is an important open problem to find the corresponding characterizations for randomized learners. In Appendix E, we provided a family of realizable instances that witnesses a super-constant gap between the optimal mistake of deterministic and randomized algorithms. This is in contrast to their classical counterparts which are always a factor of 2 within each other. One challenge (among others) of proving a tight lower bound in the randomized setting is controlling the learner's information about the agents' original features, as the adversary can no longer "look-ahead" at an algorithm's future classifiers.

---

[6]We can also derive regret bounds when $\text{OPT}_{\mathcal{H}}$ is defined based on the agents' best responses according to their own manipulation graph $G_t$ instead of $G^\star$. In this case, the regret bound would be the same up to constant factors.

**Acknowledgments.** We thank Avrim Blum for the helpful comments and discussions. This work was done while Hanrui Zhang was in residence at the Simons Laufer Mathematical Sciences Institute (formerly MSRI) in Berkeley, California, during the Fall 2023 semester. This work was supported in part by the National Science Foundation under grants DMS-1928930, CCF-2212968, and ECCS-2216899, by the Alfred P. Sloan Foundation under grant G-2021-16778, by the Simons Foundation under the Simons Collaboration on the Theory of Algorithmic Fairness, and by the Defense Advanced Research Projects Agency under cooperative agreement HR00112020003. The views expressed in this work do not necessarily reflect the position or the policy of the Government and no official endorsement should be inferred. Approved for public release; distribution is unlimited.

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

## A    Supplementary Materials for Section 2

We present the SOA algorithm in Algorithm 2.

---

**Algorithm 2:** The Classical Standard Optimal Algorithm (SOA)

**Input:** Hypothesis class $\mathcal{H}$.
**Initialization:** Version space $\mathcal{H}_0 \leftarrow \mathcal{H}$.

1 **for** $t \in [T]$ **do**
2      Observe $x_t$;
3      For $y \in \{\pm 1\}$, let $\mathcal{H}_{t-1}^{(x_t, y)} \leftarrow \{h \in \mathcal{H}_{t-1} \mid h(x_t) = \widehat{y}\}$;
4      Predict $\widehat{y}_t \leftarrow \arg\max_y \mathsf{Ldim}(\mathcal{H}_{t-1}^{(y)})$;
5      Observe $y_t$ and update version space $\mathcal{H}_t \leftarrow \mathcal{H}_{t-1}^{(x_t, y_t)}$.
6 **end**

---

## B    Supplementary Materials for Section 3

### B.1    Proof of Theorem 3.2

**Theorem 3.2** (Restated). *For any pair of hypothesis class $\mathcal{H}$ and manipulation graph $G$, any deterministic online learning algorithm $\mathcal{A}$ must suffer a mistake lower bound of* $\mathsf{Mistake}_{\mathcal{A}}(\mathcal{H}, G) \geq \mathsf{SLdim}(\mathcal{H}, G)$.

*Proof of Theorem 3.2.* Recall that to construct an adversarial sequence of agents $S = (x_t, y_t)_{t \in [d]}$ such that $\mathcal{A}$ is forced to make a mistake at every round, we will first find a path $x_1' \xrightarrow{(v_1, y_1)} x_2' \xrightarrow{(v_2, y_2)} \cdots x_d' \xrightarrow{(v_d, y_d)} x_{d+1}'$ in tree $\mathcal{T}$ which specifies the types of mistakes that the adversary wishes to induce, then reverse-engineering this path to obtain the sequence of initial feature vectors before manipulation that is realizable under $\mathcal{H}$.

**Constructing the path.** We initialize $x_1'$ to be the root of the tree $\mathcal{T}$. For all $t \leq d$ and given the history (partial path) $x_1' \xrightarrow{(v_1, y_1)} \cdots x_{t-1}' \xrightarrow{(v_{t-1}, y_{t-1})} x_t'$, we find the edge $(v_t, y_t)$ and the next node $x_{t+1}'$ as follows: run the online learning algorithm $\mathcal{A}$ for $t-1$ rounds with inputs $(v_{t'}, y_{t'})_{t' \leq t-1}$, and let $h_t$ be the outputted classifier at round $t$. We examine the labels of $h_t$ in the out-neighborhood $N_G^+[x_t']$ under graph $G$ and consider the following two cases.

    **Case 1: False negatives.** If all the feature vectors in $N_G^+[x_t']$ is labeled as negative by $h_t$, then the adversary will induce a false negative mistake by letting the post-manipulation feature vector be $x_t'$ (which is same as the original feature vector $x_t$) and the true label be positive, i.e., $(v_t, y_t) \triangleq (x_t', +1)$. We then choose the next node $x_{t+1}'$ to be the child of $x_t'$ along the false negative edge $(x_t', +1)$ in $\mathcal{T}$.

    **Case 2: False positives.** If there exists $v \in N_G^+[x_t']$ such that $h_t(v) = +1$, then the adversary will induce a false positive mistake that is observed at $v$ with true label $-1$, i.e., $(v_t, y_t) \triangleq (v, -1)$. However, we remark that the true features $x_t$ may be chosen as a different node in $N_G^-[v_t]$ to ensure realizability, which we will discuss in the reverse-engineering part. We choose $x_{t+1}'$ to be the child of $x_t'$ along the false positive edge $(v, -1)$ in $\mathcal{T}$.

**Reverse-engineering.** Repeating the above procedure for all $t \leq d$ gives us the path $x_1' \xrightarrow{(v_1, y_1)} \cdots x_d' \xrightarrow{(v_d, y_d)} x_{d+1}'$, where each $y_t$ already specifies the true labels of each agent. It remains to select the initial features $(x_t)$. Since $\mathcal{T}$ is shattered by $\mathcal{H}$, the consistency part of Definition 3.1 guarantees the existence of $h \in \mathcal{H}$ such that $\forall t < d$, there exists $x_t$ that satisfies $\widetilde{h}_G(x_t) = y_t$, where $x_t = v_t = x_t'$ if $y_t = -1$ and $x_t \in N_G^-[v_t]$ if $y_t = +1$. We let those $(x_t)_{t \in [d]}$ be agents' true feature vectors. It then follows that the sequence of agents $S = (x_t, y_t)_{t \in [d]}$ is realizable under $(\mathcal{H}, G)$ and indeed induces a mistake observed as $(v_t, y_t)$ at every round $t \in [d]$.

Finally, if $\mathsf{SLdim}(\mathcal{H}, G) < \infty$, then the above argument with $d = \mathsf{SLdim}(\mathcal{H}, G)$ proves the theorem. When $\mathsf{SLdim}(\mathcal{H}, G) = \infty$, the above argument shows $\mathsf{Mistake}_{\mathcal{A}}(\mathcal{H}, G) \geq d$ for all $d \in \mathbb{N}$, which implies that $\mathsf{Mistake}_{\mathcal{A}}(\mathcal{H}, G) = \infty$ by driving $d \to \infty$. The proof is thus complete. $\qquad\square$

## B.2 Comparison with max-degree based bounds

In this section, we consider two families of instances $(\mathcal{H}, G)$ in which our mistake bound $\mathsf{SLdim}(\mathcal{H}, G)$ from Theorem 3.3 significantly improves the previous bounds. We compare with upper bounds $O(\Delta_G^+ \cdot \log|\mathcal{H}|)$ from [Ahmadi et al., 2023] and $\tilde{O}(\Delta_G^+ \cdot \mathsf{Ldim}(\mathcal{H}))$ from [Cohen et al., 2024], where $\mathsf{Ldim}(\mathcal{H})$ denotes the classical Littlestone dimension of hypothesis class $\mathcal{H}$. We focuses on comparing with the latter bound, since it can be shown that $\mathsf{Ldim}(\mathcal{H}) \leq \log|\mathcal{H}|$ for all $\mathcal{H}$.

**Graphs with a large clique.** Our first example involves graphs with a very large clique. The main idea is that the densest part of graph may turn out to be very easy to learn since there are only a few effective hypotheses supported on it. On the other hand, the harder-to-learn part of the hypothesis class may be supported on a subgraph with a much smaller maximum degree. For this reason, the previous bounds that directly multiply the complexity of the entire graph (e.g., $\Delta_G^+$) with the complexity of the entire hypothesis class would be suboptimal.

Let $(G', \mathcal{H}')$ be a pair of manipulation graph and hypothesis class, for which we have $\mathsf{SLdim}(\mathcal{H}, G') \leq O(\Delta_{G'}^+ \cdot \mathsf{Ldim}(\mathcal{H}')$ since the strategic Littlestone dimension is a lower bound of all valid mistake bounds (Theorem 3.2). Let $N \gg \max\{|G'|, |\mathcal{H}'|\}$ be a very large integer, and $K_N$ be a clique of size $N$. We assume that the vertex set of $K_N$ is disjoint from that of $G'$. We take the hypothesis class on $K_N$ to be the set of all functions, i.e., $\{\pm 1\}^{K_N}$. Let $G = G' \cup K_N$ and $\mathcal{H} = \mathcal{H}' \times \{\pm 1\}^N$.

- **Previous bound:** Since $\Delta_G^+ = N$ and $\mathsf{Ldim}(\mathcal{H}) \geq \mathsf{Ldim}(\mathcal{H}')$. Therefore, the bound in [Cohen et al., 2024] is of order (ignoring logarithmic factors)

$$\Delta_G^+ \cdot \mathsf{Ldim}(\mathcal{H}) \geq \Omega(N \cdot \mathsf{Ldim}(\mathcal{H}')).$$

- **Our bound based on the strategic Littlestone dimension:** By Theorem 3.2, $\mathsf{SLdim}(\mathcal{H}, G)$ lower bounds the mistake bound achievable by any deterministic algorithm. Consider the following deterministic algorithm that uses two independent algorithms $\mathcal{A}_1$ and $\mathcal{A}_2$ to learn on each of the disjoint subgraphs $\mathcal{G}'$ and $K_N$. We will choose $\mathcal{A}_1$ to be the Red2Online-PMF(SOA) algorithm proposed by Cohen et al. [2024], and $\mathcal{A}_2$ be the algorithm that predicts all nodes negative until a mistake happens, at which point flips the prediction to positive on all nodes. Since the effective classifiers on $K_N$ is either all positive or all negative, $\mathcal{A}_2$ makes at most 1 mistake. We have

$$\mathsf{SLdim}(\mathcal{H}, G) \leq \mathsf{Mistake}_{\mathcal{A}_1}(\mathcal{H}', G') + \mathsf{Mistake}_{\mathcal{A}_2}(\{\pm 1\}^{K_N}, K_N) \leq \tilde{O}(\Delta_{G'}^+ \cdot \mathsf{Ldim}(\mathcal{H}')).$$

- **Improvement.** As a result, for this instance, the gap between these two algorithms is lower bounded by $N/\Delta_{G'}^+$, which can be arbitrarily large by taking $N \to \infty$.

**Random graphs in $G(n, p)$, $\mathcal{H} = \{\pm 1\}^n$.** Our second example considers random graphs $G \sim G(n, p)$, in which every (undirected) edge is realized independently with probability $p$. We will show that when $p = \omega(1/n)$ (when the random graph is "effectively" dense), with high probability over $G \sim G(n, p)$, the strategic Littlestone dimension significantly improves previous bounds.

Since $\mathcal{H}$ is extremely expressive (contains all functions), we have $\log|\mathcal{H}| = \mathsf{Ldim}(\mathcal{H}) = n$. However, we will show that even after strengthening the previous bounds by applying it on the reduced-size hypothesis class $\widetilde{\mathcal{H}}_G \subseteq \mathcal{H}$—which contains only one hypothesis in each equivalence class that induces the same effective hypothesis $\widetilde{h}_G$—the strategic Littlestone dimension $\mathsf{SLdim}(\mathcal{H}, G)$ still offers significant improvement over $\Delta_G^+ \cdot \mathsf{Ldim}(\widetilde{\mathcal{H}}_G)$.

- **Previous bound:** By concentration, $\Delta_G^+ \geq \Omega(np)$ with high probability. Moreover, with high probability, the independence number $\alpha(G)$ satisfies $\alpha(G) \geq \Omega(\log(np)/p)$ [Frieze, 1990]. Let $I(G)$ be the independent set with size $\alpha(G)$ and consider the projection of $\widetilde{\mathcal{H}}_G$ onto $I(G)$. Since there are no edges inside, the effective hypothesis coincides with the original hypothesis, therefore

$(\widetilde{\mathcal{H}}_G)_{I(G)}$ contains all functions that maps from $I(G)$ to $\{\pm 1\}$, which has classical Littlestone dimension $|I(G)| = \alpha(G)$. As a result, we have that with high probability,

$$\Delta_G^+ \cdot \mathsf{Ldim}(\widetilde{\mathcal{H}}_G) \geq \Omega(np) \cdot \Omega(\log(np)/p) \geq \Omega(n \log(np)).$$

- **Our bound based on the strategic Littlestone dimension:** Again, $\mathsf{SLdim}(\mathcal{H}, G)$ lower bounds the mistake bound of all deterministic algorithms. Consider the following algorithm: start with predicting all nodes as positive. Whenever a false positive is observed at some node $u$, flip the sign of $u$ to negative. Such an algorithm achieves a mistake bound of $n$. Therefore,

$$\mathsf{SLdim}(\mathcal{H}, G) \leq n.$$

- **Improvement** When $p = \omega(\frac{1}{n})$, with high probability, the gap between these two bounds are

$$\Omega(np) = \omega_n(1),$$

which can be made to approach $\infty$ as $n \to \infty$.

## C  Supplementary Materials for Section 4

### C.1  Algorithms for the agnostic setting

In this section, we present algorithms for the agnostic setting and prove their regret guarantees. As discussed in Section 4, the main idea behind the strategic version of agnostic-to-realizable reduction lies in our "guess" of the possible direction the original node would have come from. To do this systematically, we first need to specify a indexing system to the in-neighborhoods of every node in the graph. For each node $v \in \mathcal{X}$, we assign a unique index to each in-neighbor in $N_G^-[v]$ from the range $\{0, 1, \cdots, \Delta_G^-\}$, where $\Delta_G^-$ being the max in-degree of $G$. This indexing is specific to each $v$ and does not require consistency when indexing a common in-neighbor of different nodes. We are now ready to formally introduce our algorithms in Algorithms 3 and 4.

---

**Algorithm 3:** Agnostic Online Strategic Classification Algorithm

**Input:** Hypothesis class $\mathcal{H}$, manipulation graph $G$.
1 Let $d \leftarrow \mathsf{SLdim}(\mathcal{H}, G)$;
2 **foreach** $m \leq d, i_{1:m}, r_{1:m}$ *where* $1 \leq i_1 < \cdots < i_m \leq T, 0 \leq r_1, \cdots, r_m \leq \Delta_G^-$ **do**
3 $\quad$ Construct *Expert*$(i_{1:m}, r_{1:m})$ as in Algorithm 4.
4 **end**
5 Run *Biased Weighted Majority Vote* (Algorithm 5) on the set of experts.

---

**Algorithm 4:** *Expert*$(i_1, \cdots, i_m, r_1, \cdots, r_m; G)$

**Input:** Hypothesis class $\mathcal{H}$, manipulation graph $G$, indices for mistakes $1 \leq i_1 < \cdots < i_m \leq T$,
$\quad\quad$ indices for manipulation directions $0 \leq r_1, \cdots, r_m \leq \Delta_G^-$,
$\quad\quad$ the sequence of post-manipulation agents $(v_t, y_t)_{t \in [T]}$ received sequentially.
**Output:** Classifiers $(\hat{h}_t)_{t \in [T]}$ outputted sequentially.
**Initialization:** Simulate an instance of the SSOA algorithm with parameters $(\mathcal{H}, G)$.
1 **for** $t \in [T]$ **do**
2 $\quad$ $\hat{h}_t \leftarrow$ classifier outputted by the SSOA algorithm;
3 $\quad$ Observe the manipulated feature vector $v_t$ and the true label $y_t$;
4 $\quad$ **if** $t \in \{i_1, \cdots, i_m\}$ *(suppose $t = i_k$)* **then**
5 $\quad\quad$ $\hat{x}_t \leftarrow$ the in-neighbor in $N_G^-[v_t]$ with index $r_k$// guess of the original feature vector $x_t$
6 $\quad\quad$ $\hat{v}_t \leftarrow \mathsf{br}_{\hat{h}_t, G}(\hat{x}_t)$// simulate the post-manipulation feature vector in response to $\hat{h}_t$
7 $\quad\quad$ Update the SSOA algorithm with instance $(\hat{v}_t, -\hat{h}_t(\hat{v}_t))$.
8 $\quad$ **end**
9 **end**

---

## C.2   Proof of Lemma 4.3

**Lemma 4.3** (Restated). *For any hypothesis $h \in \mathcal{H}$ and any sequence of agents $S$, there exists an expert $\mathfrak{e}_h \in \mathfrak{E}$ that makes at most $\mathsf{SLdim}(\mathcal{H}, G)$ more mistakes than $h$.*

*Proof of Lemma 4.3.* Let us define the hypothetical sequence $S^{(h)} \triangleq (x_t, y_t^{(h)})_{t \in [T]}$, where we keep the same sequence of initial feature vectors $(x_t)$ in $S$, but adjust their labels to be $y_t^{(h)} \triangleq \widetilde{h}_G(x_t)$, i.e., the label that the effective classifier $\widetilde{h}_G$ assigns to $x_t$. Note that this sequence is defined only for analytical purpose and not required to be known by either the agnostic algorithm or the experts.

By definition, $S^{(h)}$ is realizable by the hypothesis $h \in \mathcal{H}$ under graph $G$. Therefore, Theorem 3.3 guarantees that runnning SSOA on $S^{(h)}$ gives at most $\mathsf{SLdim}(\mathcal{H}, G)$ mistakes. Let $m \le \mathsf{SLdim}(\mathcal{H}, G)$ be the number of mistakes made, and $i_1, i_2, \cdots, i_m$ be the time steps at which the mistakes occur. In addition, at every mistake $i_k = t \in [T]$, let $v_t^{(h)}$ be the post-manipulation node observed by the SSOA algorithm running on sequence $S^{(h)}$. On the other hand, let $v_t$ be the observation received by each expert. Although $v_t$ may be different from $v_t^{(h)}$ because $v_t$ is the best response to the agnostic algorithm while $v_t^{(h)}$ is the best response to SSOA, we know that $v_t$ must be an out-neighbor of $x_t$. Therefore, there must exist an index $r_k$ (where $0 \le r_k \le \Delta_G^-$) such that $x_t$ is the $r_t$-th in-neighbor of $v_t$. We argue that $Expert(i_{1:m}, r_{1:m})$ is the expert $\mathfrak{e}_h$ that we want.

We first establish the equivalence of the two following instances of SSOA:

- $\mathsf{SSOA}^{\mathfrak{e}_h}$ denotes the algorithm instance simulated by expert $\mathfrak{e}_h$;
- $\mathsf{SSOA}^h$ denotes the algorithm instance running on sequence $S^{(h)}$.

We will show by induction that both instances $\mathsf{SSOA}^{\mathfrak{e}_h}$ and $\mathsf{SSOA}^h$ have the same version space—and as a result, output the same classifier for the next round—at all time steps. This is clearly true at the base case $t = 1$, as the version spaces of both instances are initialized to be $\mathcal{H}$. Now we assume the two instances are equivalent up to $t-1$ and prove that they are still equivalent at time $t$. Since they have the same version spaces $\mathcal{H}_{t-1}$, they output the same classifiers for time step $t$. We denote this classifier by $\hat{h}_t$ as in line 2 of Algorithm 4.

If $t \notin \{i_1, \cdots, i_m\}$, then the version space $\mathcal{H}_t$ are still the same because neither instances update. Otherwise, there exists $k \in [m]$ such that $t = i_k$. Since $\mathsf{SSOA}^h$ makes a mistake at $i_k$, it will update the version space with observation $(v_t^{(h)}, y_t^{(h)}) = (v_t^{(h)}, -\hat{h}_t(v_t^{(h)}))$. On the other hand, according to line 7 of Algorithm 4, the instance $\mathsf{SSOA}^{\mathfrak{e}_h}$ is updated using observation $(\hat{v}_t, -\hat{h}_t(\hat{v}_t))$. Therefore, it suffices to show that $v_t^{(h)} = \hat{v}_t$. Since our choice of $r_k$ guarantees $x_t$ to be the $r_k$-th in-neighbor of $v_t$, we have $\hat{x}_t = x_t$ based on line 5 of Algorithm 4. Therefore, both $v_t^{(h)}$ and $\hat{v}_t$ are equal to $\mathsf{br}_{\hat{h}_t, G}(x_t)$, so they are the same. As a result, both $\mathsf{SSOA}^{\mathfrak{e}_h}$ and $\mathsf{SSOA}^h$ updates their version space using the same observation, so their $\mathcal{H}_t$ remains the same. By induction, these two instances are equivalent for all time steps.

Finally, we use the equivalence established above to prove the lemma. Using $(\hat{h}_t)_{t \in [T]}$ to denote the sequence of classifiers outputted by $\mathfrak{e}_h$, we have

$$
\mathsf{Mistake}_{\mathfrak{e}_h}(S) - \mathsf{Mistake}_h(S) = \sum_{t=1}^{T} \mathbb{1}\left\{ \hat{h}_t(\mathsf{br}_{\hat{h}_t, G}(x_t)) \ne y_t \right\} - \sum_{t=1}^{T} \mathbb{1}\left\{ h(\mathsf{br}_{h, G}(x_t)) \ne y_t \right\}
$$

$$
\le \sum_{t=1}^{T} \mathbb{1}\left\{ \hat{h}_t(\mathsf{br}_{\hat{h}_t, G}(x_t)) \ne h(\mathsf{br}_{h, G}(x_t)) \right\}
$$

$$
= \sum_{t=1}^{T} \mathbb{1}\left\{ \hat{h}_t(\mathsf{br}_{\hat{h}_t, G}(x_t)) \ne y_t^{(h)} \right\} \qquad (y_t^{(h)} = \widetilde{h}_G(x_t) \text{ in } S^{(h)})
$$

$$
= \mathsf{Mistake}_{\mathsf{SSOA}}(S^{(h)}) \qquad (\text{Equivalence of } \mathsf{SSOA}^h \text{ and } \mathsf{SSOA}^{\mathfrak{e}_h})
$$

$$
\le \mathsf{SLdim}(\mathcal{H}, G). \qquad (S^{(h)} \text{ is realizable under } \mathcal{H} \text{ and } G)
$$

The proof of the lemma is thus complete. □

## C.3 The Biased Weighted Majority Vote Algorithm

We present the algorithm in Algorithm 5.

---

**Algorithm 5:** Biased Weighted Majority Vote [Ahmadi et al., 2023]

---

**Input:** Expert class $\mathfrak{E}$, manipulation graph $G(\mathcal{X}, \mathcal{E})$ that is a supergraph of the true (unknown) manipulation graph $G^\star$.

**Initialization:** For all experts $\mathfrak{e} \in \mathfrak{E}$, set weight $w_0(\mathfrak{e}) \leftarrow 1$.

1 **for** $t \in [T]$ **do**
    /* the learner commits to a classifier $h_t$ that is constructed as follows:     */
2     **for** $v \in \mathcal{X}$ **do**
3         Let $W_t^+(v) = \sum_{\mathfrak{e} \in \mathfrak{E}: \mathfrak{e}_t(v) = +1} w_t(\mathfrak{e})$, $W_t^-(v) = \sum_{\mathfrak{e} \in \mathfrak{E}: \mathfrak{e}_t(v) = -1} w_t(\mathfrak{e})$, and
        $W_t = W_t^+(v) + W_t^-(v) = \sum_{\mathfrak{e} \in \mathfrak{E}} w_t(\mathfrak{e})$; // $\mathfrak{e}_t$ is the prediction of expert $\mathfrak{e}$ at round $t$
4         **if** $W_t^+(v) \geq W_t/(\Delta_G^+ + 2)$ **then**
5             $h_t(v) \leftarrow +1$;
6         **else**
7             $h_t(v) \leftarrow -1$;
8         **end**
9     **end**
10     Observe manipulated node $v_t$ and output prediction $h_t(v_t)$;
11     **if** $h_t(v_t) \neq y_t$ **then**
        /* If there was a mistake:     */
12         **if** $y_t = -1$ **then**
            /* False positive mistake.     */
13             $\mathfrak{E}' \leftarrow \{\mathfrak{e} \in \mathfrak{E} \mid \mathfrak{e}_t(v_t) = +1\}$; // penalize the experts that label $v_t$ as positive.
14         **else**
            /* False negative mistake.     */
15             $\hat{N}[v_t] \leftarrow N_G^+[v_t] \setminus \{x \in N_G^+[v_t], h_t(x) = +1\}$;
16             $\mathfrak{E}' \leftarrow \{\mathfrak{e} \in \mathfrak{E} \mid \forall x \in \hat{N}[v_t], \mathfrak{e}_t(x) = -1\}$;
            // penalize the experts that label all nodes in $\hat{N}[v_t]$ as negative.
17         **end**
18         if $\mathfrak{e} \in \mathfrak{E}'$, then $w_{t+1}(\mathfrak{e}) \leftarrow \gamma \cdot w_t(\mathfrak{e})$; otherwise, $w_{t+1}(\mathfrak{e}) \leftarrow w_t(\mathfrak{e})$;
19     **end**
20 **end**

---

## C.4 Proof of Theorem 4.1

**Theorem 4.1** (Restated). *For any adaptive adversarial sequence $S$ of length $T$, the Agnostic Online Strategic Classification algorithm (Algorithm 3) has regret bound*

$$\mathsf{Regret}(S, \mathcal{H}, G) \leq O\left(\Delta_G^+ \cdot \mathsf{OPT} + \Delta_G^+ \cdot \mathsf{SLdim}(\mathcal{H}, G) \cdot (\log T + \log \Delta_G^-)\right),$$

*where $\Delta_G^+$ (resp. $\Delta_G^-$) denotes the maximum out-degree (resp. in-degree) of graph $G$.*

*Proof of Theorem 4.1.* As showed in the proof sketch, combing the guarantee of Algorithm 5 and bound on $|\mathfrak{E}|$ gives

$$\mathsf{Mistake}(S, \mathcal{H}, G) \leq \Delta_G^+ \cdot \mathsf{OPT}^{\mathfrak{E}} + \Delta_G^+ \cdot \log |\mathfrak{E}| \lesssim \mathsf{OPT}^{\mathfrak{E}} + \Delta_G^+ \cdot d(\log T + \log \Delta_G^-),$$

where $\mathsf{OPT}^{\mathfrak{E}}$ denotes the optimal number of mistakes made by the best expert in $\mathfrak{E}$.

Applying Lemma 4.3 to the best hypothesis in hindsight $h^\star \in \mathcal{H}$ shows that there exists $\mathfrak{e}_{h^\star} \in \mathfrak{E}$ that makes no more than $\mathsf{OPT} + d$ mistakes, which further implies $\mathsf{OPT}^{\mathfrak{E}} \leq \mathsf{OPT} + d$. Hence, we have

$$\mathsf{Mistake}(S, \mathcal{H}, G) \lesssim \Delta_G^+ \cdot (\mathsf{OPT} + d\log T + d\log \Delta_G^-).\square$$

# D Supplementary Materials for Section 5

## D.1 Realizable graph classes

---
**Algorithm 6:** Online Strategic Classification For Relizable Graph Class

---
**Input:** Hypothesis class $\mathcal{H}$, graph class $\mathcal{G}$.

1 **foreach** $G \in \mathcal{G}$ **do**
2      Let $d_G \leftarrow \mathsf{SLdim}(\mathcal{H}, G)$;
3      **foreach** $m \leq d_G, i_{1:m}, r_{1:m}$ where $1 \leq i_1 < \cdots < i_m \leq T, 0 \leq r_1, \cdots, r_m \leq \Delta_G^-$ **do**
4          Construct $\mathit{Expert}(i_{1:m}, r_{1:m}; G)$ as in Algorithm 4.
5      **end**
6 **end**
7 Let $G_{\mathrm{union}} \leftarrow (\mathcal{X}, \sum_{G \in \mathcal{G}} \mathcal{E}_G)$ be the union of graphs in $\mathcal{G}$;
8 Run *Biased Weighted Majority Vote* (Algorithm 5) on the set of experts under graph $G_{\mathrm{union}}$.

---

**Theorem 5.1** (Restated). *For any realizable graph class $\mathcal{G}$ and any adaptive adversarial sequence $S$ of length $T$, Algorithm 6 has regret bound*

$$\mathsf{Regret}(S, \mathcal{H}, G) \leq O\left(\Delta_{\mathcal{G}}^+ \cdot \left(\mathsf{OPT}_{\mathcal{H}} + d_{\mathcal{G}} \cdot (\log T + \log \Delta_{\mathcal{G}}^-) + \log |\mathcal{G}|\right)\right),$$

*where $d_{\mathcal{G}} \triangleq \max_{G \in \mathcal{G}} \mathsf{SLdim}(\mathcal{H}, G)$ is the maximum strategic Littlestone dimension for all graphs in $\mathcal{G}$, $\Delta_{\mathcal{G}}^+ \triangleq \Delta_{G_{\mathrm{union}}}^+$ is the maximum out-degree of $G_{\mathrm{union}}$ (i.e., the union of graphs in $\mathcal{G}$), and $\Delta_{\mathcal{G}}^- \triangleq \max_{G \in \mathcal{G}} \Delta_G^-$ is the maximum max in-degree over graphs in $G$.*

*Proof of Theorem 5.1.* For each $G \in \mathcal{G}$, we use $\mathfrak{E}_G$ to denote the subset of experts constructed in Algorithm 6 for graph $G$. We also use $\mathfrak{E} \triangleq \cup_{G \in \mathcal{G}} \mathfrak{E}_G$ to denote the set of all experts.

To prove this theorem, we first revisit the regret guarantee for Algorithm 5 in Lemma D.1, especially when the input graph $G$ does not match the actual graph $G^\star$. The proof of Lemma D.1 largely follows from [Ahmadi et al., 2023], but we include it in the end of this section for completeness.

**Lemma D.1** (Regret of Algorithm 5 [Ahmadi et al., 2023]). *If Algorithm 5 is called on manipulation graph $G$ that includes all the edges in the actual manipulation graph $G^\star$, then the number of mistakes is upper bounded as follows:*

$$\mathsf{Mistake}(\mathcal{H}, G^\star) \leq O\left(\Delta_G^+ \cdot \mathsf{OPT}_{G^\star}^{\mathfrak{E}} + \Delta_G^+ \cdot \log |\mathfrak{E}|\right),$$

*where $\Delta_G^+$ is the maximum out-degree of graph $G$, and $\mathsf{OPT}_{G^\star}^{\mathfrak{E}}$ is the minimum number of mistakes made by the optimal expert under graph $G^\star$.*

Since $G_{\mathrm{union}}$ contains all edges in any $G \in \mathcal{G}$ and thus the unknown $G^\star$, Lemma D.1 that the number of mistakes made by Algorithm 6 is at most

$$\Delta_{\mathcal{G}}^+ \cdot \mathsf{OPT}_{G^\star}^{\mathfrak{E}} + \Delta_{\mathcal{G}}^+ \cdot \log |\mathfrak{E}| \leq \Delta_{\mathcal{G}}^+ \cdot \mathsf{OPT}_{G^\star}^{\mathfrak{E}} + \Delta_{\mathcal{G}}^+ \cdot \left(d_{\mathcal{G}} \cdot \log(T \Delta_{\mathcal{G}}^-) + \log |\mathcal{G}|\right),$$

where the second step uses the following upper bound on the number of experts:

$$
\begin{aligned}
|\mathfrak{E}| &\leq \sum_{G \in \mathcal{G}} \sum_{m \leq \mathsf{SLdim}(\mathcal{H}, G)} \binom{T}{m} \cdot (\Delta_G^-)^m \\
&\leq |\mathcal{G}| \cdot \sum_{m \leq d_{\mathcal{G}}} \binom{T}{m} \cdot (\Delta_{\mathcal{G}}^-)^m && (\forall G \in \mathcal{G}, \ \mathsf{SLdim}(\mathcal{H}, G) \leq d_{\mathcal{G}}, \Delta_G^- \leq \Delta_{\mathcal{G}}^-) \\
&\lesssim |\mathcal{G}| \cdot (T \cdot \Delta_{\mathcal{G}}^-)^{d_{\mathcal{G}}+1}.
\end{aligned}
$$

Therefore, it remains to show that $\mathsf{OPT}_{G^\star}^{\mathfrak{E}} \leq \mathsf{OPT}_{\mathcal{H}} + d_{\mathcal{G}}$. To this end, we apply Lemma 4.3 the expert class $\mathfrak{E}_{G^\star}$, in which all experts have the correct belief about the manipulation graph $G^\star$. For the hypothesis $h^\star \in \mathcal{H}$ that achieves $\mathsf{OPT}_{\mathcal{H}}$ under $G^\star$, there must exist $\mathfrak{e}_{h, G^\star} \in \mathfrak{E}_{G^\star} \subseteq \mathfrak{E}$ such that $\mathfrak{e}_{h, G^\star}$ makes at most $\mathsf{SLdim}(\mathcal{H}, G^\star) \leq d_{\mathcal{G}}$ more mistakes than $h^\star$ under $G^\star$. We have thus proved that $\mathsf{OPT}_{G^\star}^{\mathfrak{E}} \leq \mathsf{OPT}_{\mathcal{H}} + d_{\mathcal{G}}$, which in turn establishes the theorem. $\qquad \square$

*Proof of Lemma D.1.* Suppose a mistake is made in round $t$, we show the following claims hold:

- The total weights decrease by at least constant fraction: $W_{t+1} \leq W_t\big(1 - \gamma/(\Delta_G^+ + 2)\big)$ where $G$ is the input graph.
- The algorithm penalizes experts only if it makes a mistake on $G^\star$.

To prove these claims, we consider the following two types of mistakes.

**False positive.** Suppose $h_t(v_t)$ is positive but the true label $y_t$ is negative. According to the algorithm, $h_t$ labels $v_t$ positive only when the total weight of experts predicting positive on $v_t$ is at least $W_t/(\Delta_G^+ + 2)$. Moreover, each of their weights is decreased by a factor of $\gamma$. As a result, we have $W_{t+1} \leq W_t\big(1 - \gamma/(\Delta_G^+ + 2)\big)$ and the first claim holds.

For the second claim, note that the algorithm only penalize experts $\mathfrak{e}$ where $\mathfrak{e}_t(v_t) = +1$. Since $v_t \in N_{G^\star}^+[x_t]$, this implies $\mathfrak{e}_t(\mathrm{br}_{\mathfrak{e}_t, G^\star}(x_t)) = +1$, whereas $y_t = -1$. In other words, the experts penalized must have made a mistake under $G^\star$.

**False negative.** In the case of a false negative, the agent has not moved from a different location to $v_t$ to get classified as negative, so $v_t = x_t$. Since the agent did not move, none of the vertices in $N_{G^\star}^+[v_t]$ was labeled positive by the algorithm. However, there might exist some vertices in $N_G^+[v_t] \setminus N_{G^\star}^+[v_t]$ that are labeled as positive by the algorithm. Let $\hat{N}[v_t]$ denote the set that includes all vertices in $N_G^+[v_t]$ that are labeled as negative by $h_t$, we have

$$N_{G^\star}^+[v_t] \subseteq \hat{N}[v_t] \subseteq N_G^+[v_t].$$

According to the algorithm, for each $x \in \hat{N}[v_t]$, the total weight of experts predicting $x$ as positive is less than $W_t/(\Delta_G^+ + 2)$ where $\Delta_G^+$ is the maximum out-degree of $G$. Therefore, taking the union over all $x \in \hat{N}[v_t]$, it implies that the total weight of experts predicting negative on all $x \in \hat{N}[v_t]$ is at least

$$W_t\Big(1 - |\hat{N}[v_t]|/(\Delta_G^+ + 2)\Big) \geq W_t\Big(1 - (\Delta_G^+ + 1)/(\Delta_G^+ + 2)\Big) = W_t/(\Delta_G^+ + 2),$$

where the inequality comes from $\hat{N}[v_t] \subseteq N_G^+[v_t]$. Reducing their weights by a factor of $\gamma$ results in $W_{t+1} \leq W_t - (\gamma W_t)/(\Delta_G^+ + 2)$. The first claim holds true.

As for the second claim, if an expert $\mathfrak{e}$ is penalized, then $\mathfrak{e}_t(x) = -1$ for all $x \in \hat{N}[x_t]$. Since $N_{G^\star}^+[x_t] \subseteq \hat{N}[x_t]$, $\mathfrak{e}_t$ must label all nodes in $N_{G^\star}^+[x_t]$ as negative. In other words, $\mathfrak{e}_t(\mathrm{br}_{\mathfrak{e}_t, G^\star}(x_t)) = -1$, which means that $\mathfrak{e}$ must have made a mistake under $G^\star$. The second claim holds.

**Regret analysis.** Let $M = \mathsf{Mistake}(\mathcal{H}, G^\star)$ denote the number of mistakes made by the algorithm. Since the initial weights are all set to 1, we have $W_0 = |\mathfrak{E}|$. The first claim implies that $W_{t+1} \leq W_t\left(1 - \frac{\gamma}{\Delta_G^+ + 2}\right)$. Therefore, $W_T \leq |\mathfrak{E}|\left(1 - \frac{\gamma}{\Delta_G^+ + 2}\right)^M$.

On the other hand, we use the second claim to show that $W_T \geq \gamma^{\mathsf{OPT}_{G^\star}^{\mathfrak{E}}}$. We have proved that whenever the algorithm decreases the weight of an expert, they must have made a mistake on $G^\star$. Let $\mathfrak{e}^\star \in \mathfrak{E}$ denote the best expert that achieves the minimum number of mistakes $\mathsf{OPT}_{G^\star}^{\mathfrak{E}}$ under $G^\star$. From our argument above, the weight of $\mathfrak{e}^\star$ is penalized by no more than $\mathsf{OPT}_{G^\star}^{\mathfrak{E}}$ times. Therefore, after $T$ rounds, $W_T \geq w_T(\mathfrak{e}^\star) \geq \gamma^{\mathsf{OPT}_{G^\star}^{\mathfrak{E}}}$ where the second inequality holds since $0 < \gamma < 1$. Finally, we have:

$$\gamma^{\mathsf{OPT}_{G^\star}^{\mathfrak{E}}} \leq W_T \leq |\mathfrak{E}|\left(1 - \frac{\gamma}{\Delta_G^+ + 2}\right)^M$$

$$\Rightarrow \mathsf{OPT}_{G^\star}^{\mathfrak{E}} \cdot \ln \gamma \leq \ln|\mathfrak{E}| + M \ln\left(1 - \frac{\gamma}{\Delta_G^+ + 2}\right) \leq \ln|\mathfrak{E}| - M\frac{\gamma}{\Delta_G^+ + 2}$$

$$\Rightarrow M \leq \frac{\Delta_G^+ + 2}{\gamma} \ln|\mathfrak{E}| - \frac{\ln\gamma(\Delta_G^+ + 2)}{\gamma} \mathsf{OPT}_{G^\star}^{\mathfrak{E}}$$

By setting $\gamma = 1/e$, we bound the total number of mistakes as $M \leq e(\Delta_G^+ + 2)(\ln|\mathfrak{E}| + \mathsf{OPT}_{G^\star}^{\mathfrak{E}})$. $\quad\square$

## D.2 Agnostic graph classes

Before presenting the algorithm in the setting of agnostic graph classes, we first introduce an indexing system to the in-neighborhoods of $G_{\text{union}}$, which is constructed in the same way as described in Section 4. These indices whill be in the range $\{0, 1, \cdots, \Delta_{\mathcal{G}}^-\}$ where $\Delta_{\mathcal{G}}^-$ is the maximum in-degree of graph $G_{\text{union}}$. We now present the algorithm in Algorithms 7 and 8 and prove its regret bound in Theorem D.2.

---

**Algorithm 7:** Online Strategic Classification For Agnostic Graph Class

---

**Input:** Hypothesis class $\mathcal{H}$, graph class $\mathcal{G}$, an upper bound $N$ that satisfies $\mathsf{OPT}_{\mathcal{G}} \leq N$.

1 Let $G_{\text{union}} \leftarrow (\mathcal{X}, \sum_{G \in \mathcal{G}} \mathcal{E}_G)$ be the union of graphs in $\mathcal{G}$, $\Delta_{\mathcal{G}}^- \leftarrow \Delta_{G_{\text{union}}}^-$;

2 **foreach** $G \in \mathcal{G}$ **do**

3     Let $d_G \leftarrow \mathsf{SLdim}(\mathcal{H}, G)$;

4     **foreach** $m \leq d_G, i_{1:m} \in \binom{T}{m}, r_{1:m} \in [\Delta_G^-]^m, n \leq N, i'_{1:n} \in \binom{T}{n}, r'_{1:n} \in [\Delta_{\mathcal{G}}^-]^n$ **do**

5         Construct $Expert(i_{1:m}, r_{1:m}, i'_{1:n}, r'_{1:n}; G)$ as in Algorithm 8.

6     **end**

7 **end**

8 Run *Biased Weighted Majority Vote* (Algorithm 5) on the set of experts under graph $G_{\text{union}}$.

---

---

**Algorithm 8:** *Expert*$(i_{1:m}, r_{1:m}, i'_{1:n}, r'_{1:n}; G)$

---

**Input:** Hypothesis class $\mathcal{H}$, manipulation graph $G$, indices for mistakes $1 \leq i_1 < \cdots < i_m \leq T$, indices for manipulation directions $0 \leq r_1, \cdots, r_n \leq \Delta_G^-$, indices for imperfect graphs $1 \leq i'_1 < \cdots < i'_n \leq T$, and manipulation directions $0 \leq r'_1, \cdots, r'_n \leq \Delta_{\mathcal{G}}^-$, the sequence of post-manipulation agents $(v_t, y_t)_{t \in [T]}$ received sequentially.

**Output:** Classifiers $(\hat{h}_t)_{t \in [T]}$ outputted sequentially.

**Initialization:** Simulate an instance of the SSOA algorithm with parameters $(\mathcal{H}, G)$.

1 **for** $t \in [T]$ **do**

2     $\hat{h}_t \leftarrow$ classifier outputted by the SSOA algorithm;

3     Observe the manipulated feature vector $v_t$ and the true label $y_t$;

4     **if** $t \in \{i_1, \cdots, i_m\}$ *(suppose $t = i_k$)* **then**

        /* Guess where the original node $x_t$ comes from                              */

5         **if** $t \in \{i'_1, \cdots, i'_n\}$ *(suppose $t = i'_s$)* **then**

6             $\hat{x}_t \leftarrow$ the in-neighbor in $N_{G_{\text{union}}}^-[v_t]$ with index $r'_s$// when $G_t \neq G$, we have $G_t \subseteq G_{\text{union}}$

7         **else**

8             $\hat{x}_t \leftarrow$ the in-neighbor in $N_G^-[v_t]$ with index $r_k$// when $G_t = G$

9         **end**

10         $\hat{v}_t \leftarrow \mathsf{br}_{\hat{h}_t, G}(\hat{x}_t)$// simulate the post-manipulation feature vector in response to $\hat{h}_t$

11         Update the SSOA algorithm with instance $(\hat{v}_t, -\hat{h}_t(\hat{v}_t))$.

12     **end**

13 **end**

---

**Theorem D.2.** *For any graph class $\mathcal{G}$ and hypothesis class $\mathcal{H}$, any adaptive adversarial sequence $S$ of length $T$, and any integer $N$ that is a valid upper bound on $\mathsf{OPT}_{\mathcal{G}}$, Algorithm 7 has regret bound*

$$\mathsf{Regret}(S, \mathcal{H}, G) \leq O\left(\Delta_{\mathcal{G}}^+ \cdot \left(\mathsf{OPT}_{\mathcal{H}} + (d_{\mathcal{G}} + N) \cdot (\log T + \log \Delta_{\mathcal{G}}^-) + \log |\mathcal{G}|\right)\right),$$

*where $d_{\mathcal{G}} \triangleq \max_{G \in \mathcal{G}} \mathsf{SLdim}(\mathcal{H}, G)$ is the maximum strategic Littlestone dimension for all graphs in $\mathcal{G}$, and $\Delta_{\mathcal{G}}^+$ (resp. $\Delta_{\mathcal{G}}^-$) is the maximum out-degree (resp. in-degree) of $G_{\text{union}}$, where $G_{\text{union}}$ is the union of $\mathcal{G}$ that contains edges from all graphs in $\mathcal{G}$.*

*Proof of Theorem D.2.* Similar to the proof of Theorem 5.1, we use $\mathfrak{E}_G$ to denote the subset of experts constructed in Algorithm 7 for graph $G$, and use $\mathfrak{E} \triangleq \cup_{G \in \mathcal{G}} \mathfrak{E}_G$ to denote the set of all experts. Since we have $G_t \subseteq G_{\text{union}}$ at all time steps, Lemma D.1 on the expert set $\mathfrak{E}$ gives us an upper bound

on the number of mistakes made by Algorithm 7 as follows:
$$\mathsf{Mistake}(S, \mathcal{H}, G_{1:T}) \lesssim \Delta_{\mathcal{G}}^+ \cdot \mathsf{OPT}_{G_{1:T}}^{\mathfrak{E}} + \Delta_{\mathcal{G}}^+ \cdot \log |\mathfrak{E}|. \tag{1}$$
We establish the following lemma, which is a strengthened version of Lemma 4.3 that accounts for the possibility of $G_t \neq G$.

**Lemma D.3.** *For any hypothesis $h \in \mathcal{H}$, any sequence of agents $S$, and any sequence of graphs $G_{1:T}$ where $\sum_{t=1}^T \mathbb{1}\{N_{G_t}^+[x_t] \neq N_G^+[x_t]\} \leq N$, there exists an expert $\mathfrak{e}_h \in \mathfrak{E}_G$ (which are constructed in Algorithm 7) such that*
$$\mathsf{Mistake}_{\mathfrak{e}_h}(S, G_{1:T}) \leq \mathsf{Mistake}_h(S, G) + N + \mathsf{SLdim}(\mathcal{H}, G).$$

Applying the above lemma to $\mathfrak{E}_{G^\star}$ and using the upper bound $\mathsf{OPT}_{\mathcal{G}} = \mathbb{1}\{G_t \neq G^\star\} \leq N$, we conclude that for the optimal hypothesis $h^\star$, there must exist expert $\mathfrak{e}_{h^\star, G^\star} \in \mathfrak{E}_{G^\star} \subseteq \mathfrak{E}$, such that the number of mistakes $\mathfrak{e}_{h^\star, G^\star}$ makes under $G_{1:T}$ is at most $\mathsf{SLdim}(\mathcal{H}, G^\star) + N \leq d_{\mathcal{G}} + N$ more than that made by $h^\star$ under $G^\star$. This implies
$$\mathsf{OPT}^{\mathfrak{E}}{}_{G_{1:T}} \leq \mathsf{OPT}_{\mathcal{H}} + d_{\mathcal{G}} + N. \tag{2}$$
As for the number of experts, we have
$$|\mathfrak{E}| = \sum_{G \in \mathcal{G}} \left( \sum_{m \leq \mathsf{SLdim}(\mathcal{H}, G)} \binom{T}{m} \cdot (\Delta_G^-)^m \right) \left( \sum_{n \leq N} \binom{T}{n} \cdot (\Delta_{\mathcal{G}}^-)^n \right)$$
$$\leq |\mathcal{G}| \left( \sum_{m \leq d_{\mathcal{G}}} \binom{T}{m} \cdot (\Delta_{\mathcal{G}}^-)^m \right) \left( \sum_{n \leq N} \binom{T}{n} \cdot (\Delta_{\mathcal{G}}^-)^n \right)$$
$$\hspace{3cm} (\forall G \in \mathcal{G}, \ \mathsf{SLdim}(\mathcal{H}, G) \leq d_{\mathcal{G}}, \Delta_G^- \leq \Delta_{\mathcal{G}}^-)$$
$$\lesssim |\mathcal{G}| \cdot (T \cdot \Delta_{\mathcal{G}}^-)^{d_{\mathcal{G}}+1} \cdot (T \cdot \Delta_{\mathcal{G}}^-)^{N+1}. \tag{3}$$
Finally, plugging both (2), (3) into the bound Equation (1) gives us
$$\mathsf{Regret}(S, \mathcal{H}, G) \leq O\left(\Delta_{\mathcal{G}}^+ \cdot \left(\mathsf{OPT}_{\mathcal{H}} + (d_{\mathcal{G}} + N) \cdot (\log T + \log \Delta_{\mathcal{G}}^-) + \log |\mathcal{G}|\right)\right),$$
as desired. The proof is thus complete. $\square$

*Proof of Lemma D.3.* We use a similar approach to that of proving Lemma 4.3. We define the sequence $S^{(h)}$ and the indices $i_{1:m}, r_{1:m}$ in the same way as Lemma 4.3. In addition, we set $i'_{1:n}$ to be the time steps where $G_t \neq G$, which clearly satisfies $n \leq N$. At time step $t = i'_s$ ($s \in [k]$), since $v_t$ is the best response according to graph $G_t \subseteq G_{\mathrm{union}}$, there exists an index $r'_s \in \{0, \cdots, \Delta_{\mathcal{G}}^-\}$ such that $x_t$ is the $r'_s$-th in-neighbor under graph $G_{\mathrm{union}}$. We will show that expert $\mathfrak{e}_h = Expert(i_{1:m}, r_{1:m}, i'_{1:n}, r'_{1:n}; G)$ is the one we want.

Again, we prove this by showing the equivalence of the following two SSOA instances:

- $\mathsf{SSOA}^{\mathfrak{e}_h}$ is the algorithm instance simulated by expert $\mathfrak{e}_h$;
- $\mathsf{SSOA}^h$ is the algorithm instance running on sequence $S^{(h)}$.

Repeating the induction approach in Lemma 4.3, it suffices to show that the estimate $\hat{x}_t$ correctly matches the true $x_t$ during the rounds $i_{1:m}$. When $t \notin \{i'_1, \cdots, i'_m\}$, this is guaranteed by the way indices $r_{1:m}$ are constructed. When there exists $s \in [n]$ such that $t = i'_s$, this also holds based on the definition of $r'_{1:n}$. Therefore, the above two SSOA instances are equivalent.

Finally, we have
$$\mathsf{Mistake}_{\mathfrak{e}_h}(S, G_{1:T}) - \mathsf{Mistake}_h(S, G)$$
$$= \sum_{t=1}^T \mathbb{1}\left\{\hat{h}_t(\mathsf{br}_{\hat{h}_t, G_t}(x_t)) \neq y_t\right\} - \sum_{t=1}^T \mathbb{1}\{h(\mathsf{br}_{h,G}(x_t)) \neq y_t\}$$
$$\leq \sum_{t=1}^T \mathbb{1}\{N_{G_t}^+[x_t] \neq N_G^+[x_t]\} + \sum_{t=1}^T \mathbb{1}\left\{\hat{h}_t(\mathsf{br}_{\hat{h}_t, G}(x_t)) \neq h(\mathsf{br}_{h,G}(x_t))\right\}$$
$$\hspace{3cm} (\text{break } [T] \text{ into two parts on whether } N_{G_t}^+[x_t] \neq N_G^+[x_t])$$
$$\leq N + \mathsf{SLdim}(\mathcal{H}, G). \hspace{0.5cm} (\text{Combining } \sum_t \mathbb{1}\{G_t \neq G\} \leq N \text{ and the bound from Lemma 4.3})$$

We have thus established the lemma. $\qquad\square$

**Doubling trick to removing the assumption of knowing** $\mathsf{OPT}_{\mathcal{G}}$ We end this section with an algorithm for agnostic graph classes that does not require any prior knowledge on $\mathsf{OPT}_{\mathcal{G}}$. We present Algorithm 9 which uses Algorithm 7 as a subroutine and performs the doubling technique on the parameter $N$.

This algorithm is based on the important observation that, if $N \geq \mathsf{OPT}_{\mathcal{H}} + \mathsf{OPT}_{\mathcal{G}} + d_{\mathcal{G}} + \log|\mathcal{G}|$ (in particular this implies $N \geq \mathsf{OPT}_{\mathcal{G}}$), then Theorem D.2 guarantees that running Algorithm 7 with parameter $N$ achieves the mistake upper bound of $O\left(\Delta_{\mathcal{G}}^+ \cdot (2N) \cdot (\log T + \log \Delta_{\mathcal{G}}^-)\right)$, where the leading constant of the mistake bound can be estimated to be $\leq 4$ by a more careful analysis. Therefore, if we denote define the problem-dependent parameter $C$ to be

$$C \triangleq 8\Delta_{\mathcal{G}}^+ \cdot (\log T + \log \Delta_{\mathcal{G}}^-),$$

then as long as $N$ reaches value at least

$$N^\star \triangleq \mathsf{OPT}_{\mathcal{H}} + \mathsf{OPT}_{\mathcal{G}} + d_{\mathcal{G}} + \log|\mathcal{G}|, \tag{4}$$

running Algorithm 7 makes no more than $C \cdot N$ mistakes. Therefore, the learner just needs to estimate $N$ through the doubling trick and terminate when the observed mistake does not exceed $C$ times the estimate of $N$.

---

**Algorithm 9:** Online Strategic Classification For Agnostic Graph Class

**Input:** Hypothesis class $\mathcal{H}$, graph class $\mathcal{G}$.
1 Let $C \leftarrow 8\Delta_{\mathcal{G}}^+ \cdot (\log T + \log \Delta_{\mathcal{G}}^-)$;
**Initialization:** index of the current epoch $k \leftarrow 1$
2 **while** *total number of steps* $< T$ **do**
     /* Epoch $k$                                                           */
3    $\mathcal{A}_k \leftarrow$ a new instance of Algorithm 7 with parameters $(\mathcal{H}, \mathcal{G}, N_k = 2^k)$;
4    **while** $\mathcal{A}_k$ *has made no more than* $C \cdot 2^k$ *mistakes* **do**
5       | Run $\mathcal{A}_k$ for another round
6    **end**
     /* Exiting the while loop indicates that $N_k < N^\star$, should double estimate and enter the next epoch */
7    $k \leftarrow k + 1$
8 **end**

---

**Proposition D.4.** *For any graph class* $\mathcal{G}$, *any hypothesis class* $\mathcal{H}$, *and any sequences* $S = (x_t, y_t)_{t \in [T]}, (G_t)_{t \in [T]}$, *Algorithm 9 enjoys regret bound*

$$\mathsf{Regret}(S, \mathcal{H}, \mathcal{G}) \leq O\left(\Delta_{\mathcal{G}}^+ \cdot (\mathsf{OPT}_{\mathcal{H}} + \mathsf{OPT}_{\mathcal{G}} + d_{\mathcal{G}} + \log|\mathcal{G}|) \cdot \log(T\Delta_{\mathcal{G}}^-)\right).$$

*Proof of Proposition D.4.* Since the above argument has shown that an epoch with $N_k \geq N^\star$ will never terminate, the algorithm will have at most $k^\star = \log N^\star = \log(\mathsf{OPT}_{\mathcal{H}} + \mathsf{OPT}_{\mathcal{G}} + d_{\mathcal{G}} + \log|\mathcal{G}|)$ epochs. Moreover, since the number of mistakes made in any epoch $k$ cannot exceed $C \cdot 2^k$, the total number of mistakes (thus the regret) is upper bounded by

$$C \cdot \sum_{k=1}^{k^\star} 2^k \leq C \cdot 2^{k^\star+1} \leq O(CN^\star) \leq O\left(\Delta_{\mathcal{G}}^+ \cdot (\mathsf{OPT}_{\mathcal{H}} + \mathsf{OPT}_{\mathcal{G}} + d_{\mathcal{G}} + \log|\mathcal{G}|) \cdot \log(T\Delta_{\mathcal{G}}^-)\right).$$

This completes the proof. $\qquad\square$

## E   Randomization Gap

In this section, we present a an instance $(\mathcal{H}, G)$ in which there exists an exponential gap between the optimal mistake bound of deterministic and randomized algorithms.

Let $G$ be the star graph with center $x_0$ and $\Delta$ leaves $\{x_1, \cdots, x_\Delta\}$. $\mathcal{H} = \{h_1, \cdots, h_\Delta\}$ where $h_i(x) = \mathbb{1}\{x = x_i\}$. An adaptive adversary picks a realizable sequence $S = (x_t, y_t)_{t \in [T]}$ where each agent $(x_t, y_t)$ satisfies $y_t = \widetilde{h}^\star(x_t)$ for a fixed $h^\star = h_{i^\star} \in \mathcal{H}$. This means all the realizable choices for $(x_t, y_t)$ are restricted to the subset $\{(x_{i^\star}, +1), (x_0, +1)\} \cup \{(x_i, -1) \mid i \neq i^\star\}$.

**Deterministic algorithms**  If the learner is restricted to using deterministic algorithms, then Ahmadi et al. [2023, Theorem 4.6] showed that the optimal mistake is lower bounded by $\Delta - 1$. This implies $\mathcal{M}^{\mathrm{det}}(\mathcal{H}, G) \geq \Delta - 1$.

**Randomized algorithms**  If the learner is allowed to use randomness, we will construct an randomized algorithm $\mathcal{A}$ that enjoys an expected mistake bound of $\log \Delta$. This would imply $\mathcal{M}^{\mathrm{rand}}(\mathcal{H}, G) \leq \log \Delta$, which in turn witnesses a super-constant gap between the minmax optimal mistake bounds for deterministic and randomized learners, i.e.,

$$\frac{\mathcal{M}^{\mathrm{det}}(\mathcal{H}, G)}{\mathcal{M}^{\mathrm{rand}}(\mathcal{H}, G)} \geq \omega(1).$$

The algorithm $\mathcal{A}$ maintains a version space $\mathcal{H}_t$ that consists of all the hypotheses consistent with the history up to time $t$. This version space is initialized to be $\mathcal{H}_1 \leftarrow \mathcal{H}$. At every round, the learner commits to a distribution over classifiers that randomly pick a classifier from the version space, i.e., $h_t \sim \mathsf{Unif}(\mathcal{H}_t)$. Let $M(k)$ be the expected of mistakes by $\mathcal{A}$ in the future starting from time step $t$ where $|\mathcal{H}_t| = k$. Consider the following cases:

- If the adversary chooses $(x_t, y_t) = (x_{i^\star}, +1)$, then the learner makes a mistake with probability $1 - \frac{1}{k}$, but gets to know $i^\star$ afterwards and will make no more mistakes in the future. In this case, we have $M(k) = 1 - \frac{1}{k}$.

- If the adversary chooses $(x_t, y_t) = (x_0, +1)$, then the learner never makes mistakes, but also gains no information. The version space remains the same ($\mathcal{H}_{t+1} = \mathcal{H}_t$). The expected mistakes in the future is still $M(k)$.

- If the adversary chooses $(x_t, y_t) = (x_i, -1)$ for some $i \neq i^\star$, then the learner makes a mistake only when $h_t = h_i$, which happens with probability $\frac{1}{k}$. On the other hand, no matter which $h_t$ gets realized, the learner can always observe $(v_t, y_t) = (x_i, -1)$ and update the version space to be $\mathcal{H}_{t+1} = \mathcal{H}_t \setminus \{h_i\}$. By symmetry of the star graph, the expected mistakes for future rounds is $M(k-1)$. We thus have $M(k) = \frac{1}{k} + M(k-1)$.

Overall, we have

$$M(k) \leq \max \left\{ 1 - \frac{1}{k}, \ M(k), \ \frac{1}{k} + M(k-1) \right\}.$$

Solving the above recurrence relation gives us $M(k) \leq \log k$, which implies $\mathcal{M}^{\mathrm{rand}}(\mathcal{H}, G) \leq \mathsf{Mistake}_{\mathcal{A}}(\mathcal{H}, G) = M(\Delta) \leq \log \Delta$.

