# OpenReview forum: "Strategic Littlestone Dimension: Improved Bounds on Online Strategic Classification"
_NeurIPS.cc/2024/Conference — NeurIPS 2024 poster_

### Official Review · Reviewer_6Ry7 · 2024-07-07

**Soundness:** 3
**Presentation:** 3
**Contribution:** 3
**Rating:** 6
**Confidence:** 3

**Summary:**

This paper studies online strategic classification from a combinatorial perspective. The paper defines a new combinatorial dimension called the Strategic Littlestone dimension that jointly captures the complexity of the hypothesis class and the manipulation graph. They show that the Strategic Littlestone dimension exactly quantifies the minimax number of mistakes made by deterministic learners in the realize setting. The paper also provides improved upper bounds in the agnostic setting by modifying the classic agnostic-to-realizable reduction to handle the fact that the learner does not observe true features. Finally, the paper considers the case where the manipulation graph is not known to the learner but belongs to a family of graphs which is known to the learner. They provide bounds on the minimax value in both the realizable and agnostic settings in this case.

**Strengths:**

- The paper is well-written and easy to follow
- The problem setting is well-motivated
- The technical contributions are novel and improve upon existing results. In particular, I found the proof of the lowerbound in terms of the strategic Littlestone dimension to be nice.

**Weaknesses:**

- Lack of lower bounds. Apart from the known manipulation graph, realizable setting (Theorem 3.2), lower bounds on the minimax regret in terms of the strategic Littlestone dimension are not provided.

- Lack of results for randomized learners in the realizable setting. While the authors do discuss randomized learnability, and its difficulty in the discussion section, I find it a bit unsatisfying. Without lower bounds for randomized learners, it is not clear whether the strategic Littlestone dimension characterizes realizable strategic online learnability in full generality. Meaning,  as far as I can tell, there could be a separation between deterministic and randomized realizable learnability for strategic online classification (correct me if I'm wrong here).

- Deterministic learners in the agnostic setting. It is well known that in order to achieve sub linear regret in the traditional online classification setting, one needs randomization in the agnostic setting. Thus, it is a bit strange to me that the authors study/construct deterministic learners for strategic online classification in the agnostic setting. This also leaves open the question of whether the strategic Littlestone dimension characterizes agnostic online learnability.

- The two points above bring into question the utility/usefulness of the strategic Littlestone dimension. Compared to say the Littlestone dimension, which not only qualitatively characterizes online learnability in the realizable and agnostic settings but also exactly quantifies the minimax rates, the same (at least as of right now), cannot be said of the strategic Littlestone dimension. As far as I can tell, the strategic Littletone dimension only characterizes deterministic realizable learnability.

- Lack of intuition behind the dimension. In traditional online classification, the intuition behind needing shattered Littlestone trees is very easy from a lower bound perspective  - we just need to tell the adversary what to play for every move of the learner. However, this sort of logic does not seem to work for strategic Littlestone trees due to the need for reverse engineering. Unfortunately, I did not fully understand the intuition behind the strategic Littlestone tree from the proof sketch of the lower bound in the main text.  I had to read the full proof in the Appendix to fully understanding the structure of the strategic Littlestone tree. I think the paper can benefit from a more detailed discussion about the differences between shattered Littlestone trees and shattered strategic Littlestone trees and how the adversary should use strategic Littlestone trees to construct hard streams. In addition, I think to improve intuition, it would be nice to buff up the proof sketch of Theorem 3.2, and even better to include the full proof in the main text.

- Minor: there is a typo in the second bullet point in Def. 3.1. I think it should be $N_G^{-}$.

**Questions:**

-  In the case of unknown manipulations graphs, is the finiteness of the graph class necessary? Is there a joint dimension of the graph class and the hypothesis class that captures learnability and the minimax rates in this setting?

- Your definition of realizability states that if $(x_1, y_1), ..., (x_T, y_T)$ is the sequence of agents chosen by the adversary, there is a hypothesis $h \in H$ such that $h(br_{G, h}(x_t)) = y_t$. That is, the label $y_t$ agrees with the value of $h$ on the agents best-response to $x_t$ w.r.t $h$. Is this the most natural definition of realizability? What about simply requiring that $h(x_t) = y_t$ for all $t \in [T]$. Then, perhaps the fact that the learner observes $v_t$ instead of $x_t$ but still needs to correctly predict $y_t$ can be viewed as the learner observing noisy features? It would be nice if you can comment about why you choose this particular notion of regret and realizability.

- Can you comment about any known separations between deterministic and randomized learnability for strategic online classification in the realizable setting?

**Limitations:**

No limitations.

---

> ### Author Rebuttal · Authors · 2024-08-07
>
> Thank you for the detailed and insightful comments.
>
> > Deterministic learners in the agnostic setting
>
> Our focus on deterministic algorithms is motivated by real-world applications of strategic classification where legal or regulatory requirements often demand the learner to use deterministic algorithms. For example, in college admissions, institutions are required or expected to publish clear and transparent decision criteria, as randomization could be seen as arbitrary or unfair, weakening the trustworthiness of the admissions process.
>
> At the technical level, our construction of representative experts (Lemma 4.3) relies on a careful coupling analysis that applies to any trajectory of classifiers chosen by the learner, regardless of whether randomness is used or not. However, for the strategic variant of the learning from expert advice algorithm, the only known agnostic algorithm (from [Ahmadi et al., EC’23]) is also deterministic, which accounts for the deterministic nature of our agnostic algorithm. Therefore, to extend this technique to randomized algorithms, one key challenge is to design randomized agnostic algorithms in the finite expert setting with at most logarithmic dependence on the size of the experts. We believe this is an interesting and non-trivial question for future research.
>
> > Finiteness of the graph class in the unknown manipulation graph setting, and joint dimension of the graph class and the hypothesis class
>
> Thank you for the insightful question. While the finiteness of the graph class is necessary in the worst case scenario (see [Prop 14, Cohen et al, 2024] for a lower bound in terms of $\log|\mathcal{G}|$), it is not necessary for every instance. We agree that introducing a joint complexity measure for the graph class and the hypothesis class to characterize minmax rates is a valuable open question. Some technical challenges we encountered in extending our single-graph SLDim to the multi-graph setting include:
>
> - In the structure of the strategic Littlestone tree (see Lines 247-248), the set of outgoing edges from a node depends on its out-neighborhood under the manipulation graph. When dealing with a family of graphs, one might attempt to use the union of the out-neighborhood for each graph in the class. However, that causes the tree to be too wide and shallow, resulting in a dimension that is smaller than the actual minmax mistake bound. In particular, it cannot become a valid upper bound because the branches no longer reflect the true types of mistakes that the learner can possibly make.
>
> - If one keeps track of a “graph version space” and to determine the structure of the Littlestone tree, there may be a monotonicity issue. As graphs are eliminated from the graph version space, the set of feasible manipulations shrinks, which may potentially cause the tree to become thinner and deeper, thus increasing the dimension of the subgraph! This is in contrast to the monotonicity in terms of the hypothesis class which is less subtle and almost immediate from definition.
>
> > Definition of realizability and regret
>
> Thanks for the question. We believe this is an important point and we will add a remark to future versions of our paper. Unlike the traditional notion of realizability that simply assumes $h(x_t)=y_t$ some $h\in H$ across all $t\in[T]$, we use the strategic notion of realizability that requires $h(BR_{G,h}(x_t))=y_t$. In other words, we require that a hypothesis $h\in H$ classifies each agent correctly if the agents also best respond to $h$. We adopt this notion for several reasons:
>
> - This notion is rooted in the concepts of Stackelberg value and Stackelberg regret in game theory, which account for agents’ best responses and serve as a natural benchmark in strategic settings.
>
> - It places the learner and the optimal hypothesis on equal footing when evaluating the number of mistakes they make, unlike the standard notion of realizability which implicitly assumes that agents manipulate against the learner but not against the optimal hypothesis.
>
> - Under this strategic notion of realizability, there always exists a learning algorithm in hindsight that can achieve zero mistakes against a realizable sequence, such as when the algorithm implements the optimal hypothesis throughout. This aligns with the purpose of introducing realizability in the first place. In contrast, it is unclear that any algorithm would be able to achieve zero mistakes against a standard realizable sequence. Previous work has shown strong incompatibility between these two notions in the context of linear strategic classification setting [Chen, Liu & Podimata, NeurIPS’20].
>
> > Separations between deterministic and randomized learnability
>
> Yes, we showed a separation between deterministic and randomized mistake bounds in Appendix E. Specifically, we constructed a family of instances where for each value of $\Delta$, there exists an instance where the minmax bound for deterministic algorithms is at least $\Delta-1$, but the minmax bound for randomized algorithms is at most $\log\Delta$. This class witnesses a super-constant gap between deterministic and randomized bounds, unlike their non-strategic counterparts which only differ by a factor of 2. This implies that the proposed SLDim does not characterize randomized learnability and that characterizing learnability in the randomized setting is highly nontrivial.

---

> > ### Comment · Reviewer_6Ry7 · 2024-08-07
> >
> > Thanks for your response.

---

### Official Review · Reviewer_ZNn3 · 2024-07-08

**Soundness:** 3
**Presentation:** 2
**Contribution:** 3
**Rating:** 6
**Confidence:** 3

**Summary:**

This paper studies an adversarial online setting where the agents can manipulate their feature vector $x_t$ to some feature vector $x'_t$ given a graph of manipulation rules $G$. The learner observes only $x'_t$ and knows in advance $G$.
The usual goal is to obtain sublinear regret, with respect to some hypothesis class. In the realizable setting, this is equivalent to minimizing the number of mistakes, also known as the "mistake bound model".
When $G$ is "empty", meaning there are no edges, we get the standard online model.

The main contributions are as follows:

In the realizable setting, the authors define a Littlestone tree that incorporates the manipulations and defines a notion of the Littlestone dimension for these trees. Extending to the agnostic setting, the authors use a known technique for constructing a finite "online cover" of the hypothesis class, and then apply an algorithm for the finite case (version of multiplicative weights update).

Moreover, an extended model is introduced, where there is a finite set of manipulation graphs from which a graph is chosen. When all agents use the same graph from this set this is called the realizable case otherwise it is an agnostic setting.

**Strengths:**

The results in this paper improve upon previous papers that study this question.

The techniques are clear, and the proofs are correct (as far as I checked).

**Weaknesses:**

My main concern about the paper is the lack of technical novelty, ideas from standard online learning translate pretty smoothly to the strategic online learning setting.

The definition of the Littlestone tree is straightforward, taking the "strategic loss" into account. Indeed, the upper and lower bound proofs are very similar to those of the standard Littlestone dimension and the SOA algorithm.

The ideas in the agnostic setting are also quite standard: the agnostic online learning technique of constructing a finite "online cover" for the class (by Ben-David et al.) and using a version of multiplicative weights on the finite class (by Ahmadi et al.).

What is the technical contribution of this paper?

The writing of the paper could be improved. I understand why the proofs should go through, but some definitions are really confusing.
See the next section.

**Questions:**

There are many parts of the paper where the writing/definitions are confusing.

Paragraph on the manipulation rule: $h$ denotes the classifier or the loss function? Why does it make sense to define the best response on points where $h$ returns $1$? I don't see how it compiles if it's not the loss function.
It's not clear if it is consistent with lines 230-235: how do you define false positive and false negative? with respect to $h$ or the loss of $h$?

Line 71 ״under the respective particular parametrization״ what does it mean?
I understand what the line afterward means, but line 71 is not clear to me.

Lines 94-95: "First, to construct the set of representative challenges". Is it a typo?

Minor:

Line 34: "decision maker has little or no prior knowledge about the population being classified", this is also the case where the learner has no prior knowledge of the distribution.

Sometimes the learner is also referred to as the decision maker (mostly in the intro), it's worth making it clear.

**Limitations:**

The limitations are addressed properly.

---

> ### Author Rebuttal · Authors · 2024-08-07
>
> Thank you for your the detailed review and valuable feedback.
>
> > Technical contribution
>
> We respectfully disagree that our paper lacks technical novelty because it applies ideas from standard online learning. While we build on established methodologies such as the Littlestone tree, SOA learning rules, and the agnostic-to-realizable reduction via experts, these are foundational philosophies in learning theory literature that require different technical insights when applied to different settings. An active and expanding line of work have also been applying these methodologies to understand the learnability and optimal rates in various contexts, such as the multiclass Littlestone dimension [Daniely et al., COLT’11], Natarajan-Littlestone dimension [Kalavasis et al., NeurIPS’22], the randomized Littlestone dimension [Filmus et al., COLT’23], the VC-Littlestone tree [Bousquet et al., STOC’21], to name a few. While these results share common philosophies, each presents unique technical challenges and requires new insights.
>
> In the strategic classification context, one significant challenge for us is the information asymmetry between the learner and adversary regarding the true features before manipulation. This challenge is not addressed by previous works on non-strategic learning where the true features are always observable. Below we outline our main technical contributions in addressing these challenges:
>
> - **Structural insights.** As discussed in Lines 225-242 of Section 3, our construction of the strategic Littlestone tree accounts for the asymmetric information carried by false positives and false negatives by creating an asymmetric number of branches. We also introduce a carefully designed consistency rule to ensure realizability of the adversary’s choices.
>
> - **Lower bounds:** While constructing a sequence of post-manipulation observations that the adversary wishes to induce to the learner is straightforward, a critical challenge is in finding a realizable sequence of initial features before manipulation. We address this by combining our new consistency rule with a novel reverse-engineering technique.
>
> - **Upper bounds:** As discussed in Lines 285-290, the main challenge in proving the upper bound is to construct a classification rule that simultaneously satisfies the “progress on mistakes” property for all features in the space. Since each layer of the SL tree inspects the entire neighborhood of a certain node, one cannot optimize each node independently as in non-strategic settings. We resolve this by favoring positive labels whenever false positives decrease the SLDim, a technique also rooted in the asymmetric information carried by false positives vs negatives.
>
> - **Agnostic reduction:** As discussed in Lines 88-93, our expert set needs to effectively guess the direction from which each agent moves from. We also extend this technique to the unknown graph setting.
> We will add a more detailed discussion about these technical challenges and contributions in the revisions of our paper.
>
> > Manipulation rule, false positives and false negatives
>
> Implicit in our model is that all agents prefer positive labels over negative labels. While we use $h$ to denote the learner’s choice of classifier, you’re correct that agents’ loss function can also be written in terms of $h$. Specifically, under classifier $h$, an agent with true features $x$ and post-manipulation features $v$ incurs a loss of $-h(v)-\infty\cdot\mathbf{1}\{(x,v)\not\in E\}$, where the second term is to ensure feasibility of manipulation from $x$ to $v$. To minimize this loss, agents will choose $v$ such that $h(v)=+1$ if possible. That’s why we define the best response on nodes where $h$ return $1$. We will clarify this assumption explicitly in future versions.
>
> If an agent $(x,y)$’s true label $y$ is negative but she manipulates her features to receive a positive classification $h(v)=+1$, the learner has made a “false-positive mistake”. In this case, the observation available to the learner is represented as the pair (agent’s observable features, agent’s true label)$=(v,-1)$. Conversely, if $y=+1$ but no neighbor of $x$ is classified as positive by $h$, the agent will not manipulate and will receive a negative label $h(x)=-1$, resulting in a “false-negative mistake”. The learner’s observations are denoted by the pair  (agent’s observable features, agent’s true label)$=(x,+1)$.
>
> > Line 71 ״under the respective particular parametrization״
>
> Thank you for pointing this out, we are sorry for the confusion. What we meant is that the (near)-optimality of the mistake bounds in previous work [Cohen et al., Ahmadi et al.] are established under the assumption that the mistake bound must be parametrized as a function on the Littlestone dimension and/or the graph’s out-degree. These works show optimality in a specific instance in which the bounds cannot be improved when parameterized in these particular ways. However, these bounds are not tight for all instances, especially when the parameters such as LDim or out-degree are infinite. We will clarify this in more detail.
>
> > Typo in Lines 94-95
>
> Yes, this is a typo. We meant to write “to construct the set of representative experts”.
>
> > Learner’s prior knowledge
>
> While it is true that in the stochastic/offline setting with an unknown distribution, the learner has no prior knowledge about the population being classified, we want to emphasize that this is still easier than our online setting because even though the distribution is unknown, the learner knows that that there exists an underlying distribution, allowing her to learn a good classifier by estimating the distribution. In contrast, in the online setting, agents are adversarially chosen on-the-fly and do not come from any prespecified distribution.
>
> > Learner vs decision maker
>
> Yes, we will make it clear in future versions of our paper.

---

> > ### Comment · Reviewer_ZNn3 · 2024-08-12
> > **Raising my score**
> >
> > Thanks for your response.
> > I believe the contribution is sufficient for acceptance.
> >
> > Please make the relevant changes in the presentation so it will be easier to understand the main definitions.

---

> > > ### Author Response · Authors · 2024-08-14
> > >
> > > Thank you for raising the score! We will incorporate your suggestions into revisions of the paper to make the presentation more clear.

---

### Official Review · Reviewer_Mb38 · 2024-07-10

**Soundness:** 4
**Presentation:** 4
**Contribution:** 2
**Rating:** 7
**Confidence:** 3

**Summary:**

The authors continue the study of strategic classification in the online setting, where an agent can manipulate the instance features to potentially force a positive prediction (governed by a manipulation graph). In the *realizable* case, the authors provide a strategic variant of the Littlestone dimension and the SOA algorithm yielding the exact instance-dependent mistake rate. This improves upon degree-dependent mistake bounds of previous work. They also consider two individual additional directions: the agnostic case and the case when the manipulation graph is unknown. In the first they improve upon existing regret bounds, while in the latter they state novel upper bounds almost matching existing lower bounds.

**Strengths:**

Well written, interesting setting, and natural continuation ob previous work.
Tight characterization of the realizable setting.

--- rebuttal ---
updated from 6 to 7

**Weaknesses:**

Strength of the agnostic result is a bit unclear / open (see questions / limitations below).

**Questions:**

How does Thm 4.1. (the agnostic regret bound) relate to Prop. 30 in Cohen et al. [2024]? Moreover, are there somewhat tight (say up to log factors) lower bounds on the agnostic regret bound? In particular is the $\mathcal{O}(\Delta\cdot\mathrm{SLdim})$ term necessary in the upper bound (Thm 4.1)? (From your remark I understood that only $\Delta\cdot\mathrm{OPT}$ and $\mathrm{SLdim}$ are known lower bounds).

Is the assumption that $X$ is discrete a strong one/necessary? We can just model the "manipulation graph" on an arbitrary domain $X$ as a function $f:X\to P(X)$, where $f(x)$ is the set of allowed manipulations of $x$. Are there any difficulties generalizing here?

Is a "symmetric" setting possible where the agent can (adversarially) modify $x$ to $v\in N[x]$ no matter the label $h(x)$?. The motivation could be that some agents want to be strategically classified as $-1$ while others as $1$.

**Limitations:**

The achieved agnostic regret bound is not compared to the agnostic one in Cohen et al. [2024, Prop 30], or perhaps I missed it, see question above. Please clarify as this would help to judge the novelty and strength of the result.

---

> ### Author Rebuttal · Authors · 2024-08-07
>
> Thank you for the positive feedback and the insightful comments.
>
> > Comparison of our agnostic regret bound (Thm 4.1) to that of Cohen et al. (Prop 30), and whether there are tight lower bounds
>
> Thank you for your question. Our agnostic algorithm that achieves the mistake bound in Thm 4.1 only requires observing post-manipulation features after committing to a classifier at each round. In contrast, Prop 30 of Cohen et al. requires the learner to observe pre-manipulation data before choosing a classifier and then observe post-manipulation data afterwards. This makes their setting strictly easier than ours. We will add a remark about this comparison to revised versions of our paper.
>
> About lower bounds in the agnostic setting, we agree that both $\Delta\cdot\text{OPT}$ and $\text{SLDim}$ are valid lower bounds but it’s unclear whether the $\Delta\cdot\text{SLDim}$ term is necessary. It remains an important open question to derive lower bounds for this agnostic setting.
>
> > Does the domain $\mathcal{X}$ need to be discrete?
>
> No, the assumption on the discrete domain $\mathcal{X}$ is not necessary. We totally agree that our results can be immediately extended to any domain with arbitrary predefined manipulations characterized by some abstract function $f$. We will clarify this in revisions.
>
> > Symmetric setting where agents can adversarially modify their features to the neighboring features
>
> We agree that your proposed setting is also well-motivated, though it takes a different approach compared to our work. Our focus is on strategic modifications, which can be viewed as a special case of the fully adversarial setting that you mentioned. The setting you described–where each $x$ can adversarially move to any neighbor in $N(x)$ regardless of the label–has also been explored in the adversarial robustness literature, particularly in the context of offline learning where dat appoints are sampled from a fixed distribution (see, eg, [Montasser et al., NeurIPS’22]). Another related model (also in the offline setting) that reflects your motivation is proposed by [Sundaram et al., ICML’21] which introduces another parameter $r\in\mathbb{R}$ that indicates how much a data point prefers label $+1$ over $-1$. In this model, a negative $r$ implies that some agent wants to be classified as $-1$ rather than $+1$. Although our current approach does not generalize immediately, we believe that it is an interesting direction to bring either model to the online learning setting.

---

> > ### Comment · Reviewer_Mb38 · 2024-08-12
> >
> > Thanks for the clarification, I raised my score. This is a timely paper studying an important problem.

---

> > > ### Author Response · Authors · 2024-08-14
> > >
> > > Thank you for raising the score!

---

### Official Review · Reviewer_fR1n · 2024-07-12

**Soundness:** 2
**Presentation:** 2
**Contribution:** 3
**Rating:** 5
**Confidence:** 2

**Summary:**

The paper tackles online binary classification where agents manipulate observable features for positive outcomes. It introduces the Strategic Littlestone Dimension (SLD), a new measure capturing the complexity of the hypothesis class and manipulation graph, demonstrating its role in achieving optimal mistake bounds for deterministic algorithms in the realizable setting. It also improves regret bounds in the agnostic setting by refining agnostic-to-realizable reductions and addressing unobserved original features. Additionally, it relaxes the assumption of a known manipulation graph, deriving regret bounds for scenarios with imperfect graph knowledge in both realizable and agnostic settings.

**Strengths:**

The problem is well-motivated and interesting. The paper is also well-structured. The result based on the notion of the Strategic Littlestone Dimension is tight, with matched lower and upper bounds. Although I did not check the proof, the results of this paper appear to be correct.

**Weaknesses:**

My main concern is twofold:

1. The paper primarily concerns deterministic learning algorithms, while most robust algorithms dealing with adversaries are stochastic, which greatly limits the practical relevance of the paper.

2. The computational complexity of Algorithm 1 is not clear. Line 3 of Algorithm 1 looks computationally expensive. Can authors comment on the computational complexity of the algorithm?

**Questions:**

See weaknesses.

**Limitations:**

The authors did discuss the limitations of their work.

---

> ### Author Rebuttal · Authors · 2024-08-07
>
> Thank you for your questions and comments.
>
> > Weakness 1: deterministic learning algorithms
>
> While we agree that many robust algorithms rely on randomization to deal with adversaries, we want to highlight that in the context of strategic classification, there are important scenarios where the learner must use deterministic algorithms due to legal or regulatory requirements. An example is college admissions, where institutions are required to publish clear and transparent decision criteria (aka classifiers), as randomization could be perceived as arbitrary or unfair, weakening the trustworthiness of the admissions process. This motivates our study of deterministic algorithms and highlights its practical relevance. We will add this note to the revised paper.
>
> In addition, characterizing the minmax rates for randomized algorithms is highly nontrivial. In Appendix E, we have constructed instances for all $\Delta$ where the deterministic minmax bound is $\Omega(\Delta)$ but the randomized minmax bound is $O(\log\Delta)$. This reveals a super constant gap that did not exist in the non-strategic setting.
>
> On the technical front, one of the many challenges of establishing a lower bound for randomized algorithms is that the adversary's ability to reverse-engineer the post-manipulation features to obtain pre-manipulation ones relies on “looking ahead” at the learner’s algorithm. However, if the learner uses randomness, the adversary cannot fully control which mistakes will occur or what information they provide to the learner. This challenge is deeply rooted in the information asymmetry inherent in the strategic setting.
>
> > Weakness 2: computational complexity of Algorithm 1
>
> While we acknowledge that Algorithm 1 can be computationally expensive to implement, we want to remark that our primary focus is on the *statistical complexity* of learning in the presence of strategic manipulations, rather than the computational complexity. It is often the case that statistically optimal algorithms become computationally intensive — this is also true in the traditional (non-strategic) setting, where the SOA algorithm that enjoys minimax optimal mistake bound is also computationally expensive, and there are even computational hardness results for learning certain classes [Hazan and Koren, STOC’16]. We believe that exploring the tradeoff between computational and statistical complexity is an interesting direction for future research.

---

> > ### Comment · Reviewer_fR1n · 2024-08-11
> >
> > Many thanks for your response.

---

### Official Review · Reviewer_wZa2 · 2024-07-13

**Soundness:** 4
**Presentation:** 4
**Contribution:** 4
**Rating:** 8
**Confidence:** 3

**Summary:**

This paper considers the problem of online binary classification when each data point can strategically manipulate its features in a discrete way which is captured by a manipulation graph.

Protocol and regret: In each round, the learner picks a deterministic classifier $h_t$, and then the data point $(x_t,y_t)$ arrives, and manipulates its feature according to graph $G$ from $x_t$ to $v_t$. The learner receives only $v_t$ and incurs loss $\mathbb{1}[h_t(v_t) \neq y_t]$. The regret is measured with respect to the best fixed $h \in \mathcal{H}$.

**contributions**


1. They introduced an elegant notion of Strategic Littlestone Dimension $\mathrm{SLdim}(\mathcal{H}, G)$, which is based on hypothesis class $\mathcal{H}$ and manipulation graph $G$

2. They showed that $\mathrm{SLdim}(\mathcal{H}, G)$ fully characterizes the optimal mistake bound for a realizable case, by showing a lower bound on the mistake using a strategic counterpart of shattering tree and an upper bound using a strategic version of SOA.  (Note: The realizable case here means there exists some $h \in \mathcal{H}$ with no mistakes on the manipulated data. so $h$ does not necessarily need to correctly classify the actual data.)
3. They extended their results for the agnostic case and showed an improved upper bound compared to previous works. They used the idea of approximating the whole class $\mathcal{H}$ using a finite set of representative experts. (Their benchmark is $\Delta_G^+ \cdot OPT$, where $OPT$ is the error of the best hypothesis and $\Delta_G^+$ is maximum out-degree of graph $G$)
4. Finally, they designed algorithms with regret upper bound for the setting where the manipulation graph is unknown to the learner. They consider two cases: (i) all data points use the same unknown manipulation graph from a set of graphs, (ii) each data point uses a separate manipulation graph (case (ii) better captures those real-world applications in which different data points might have different capabilities for manipulation.)

**Strengths:**

This is a very strong paper

- Wring is clear easy to follow and well cited. Math notations and technical proofs are also clean and clear.
- Significance: the notion of strategic Littlestone dimension: This is a very valuable extension of the Littlestone dimension which characterizes the optimal mistake-bound --> it improves our understanding of the complexity of online binary classification with the presence of strategic behavior
- Extension for unknown manipulation graph is valuable and I suspect further research by a broad range of researchers in that line
- Some of the previous results relied on knowledge about both pre-manipulation data and post-manipulation data (e.g. [Cohen et al. 2024]) but this paper uses only post-manipulation data
- A detailed comparison with previous works in the appendix for the realizable case (I didn't closely check all the details)

**Weaknesses:**

I don't find any major weakness at all.

**Questions:**

1. In line 101, you mentioned that, your approach yields an improved bound for the agnostic setting. Can you please provide more details about this improvement? I believe you are comparing the upper bound in Theorem 4.1. with an upper bound from previous work.
2. In your setting, you assumed that each strategic data point only gets manipulated for a positive classification and if no manipulation leads to a positive classification, then the data point remains unchanged. Is this a necessary assumption for your results? In other words, I was wondering, do you anticipate that your result can be easily extended to the setting where all data points (even those without a chance of getting positive classification)  have the potential to be manipulated?

**Limitations:**

As mentioned by the authors in the checklist, this paper is theoretical with no direct implications. I completely agree with this.

However, I want to note that the motivation for strategic classification comes from real-world problems. I think the progress toward this direction by theoretical researchers and practitioners can only be beneficial for society. This paper improves our theoretical understanding of the complexity of this problem in a somewhat stylized case.

I can think of only one possible potential limitation (in a very high-level sense). Suppose a practitioner decided to use manipulation robust learning algorithms in their specific application. They need to model a set of manipulation graphs $\mathcal{G}$. If due to inaccuracy in the modelling by the practitioner, $\mathcal{G}$ only captures the certain type of manipulations done by group $A$ and does not capture the type of manipulations done by group $B$, then the resulting learning algorithm will be only robust with respect to a certain type of manipulation from group $A$. Therefore, group $B$ **might** have gained some unfair advantages. Hence, it is important to model the manipulation graph set $\mathcal{G}$ accurately.

---

> ### Author Rebuttal · Authors · 2024-08-07
>
> Thank you for the positive feedback on our work. We are happy to see that you find our notion of Strategic Littlestone Dimension elegant and valuable.
>
> > Question 1: Improved bound for the agnostic setting
>
> This improvement results from the comparison between Theorem 4.1 in our paper and the agnostic mistake bound in [Theorem 4.5, Ahmadi et al., 2023]. Both papers study the same setting of agnostic online strategic classification and design deterministic algorithms that achieve mistake bounds in terms of $\Delta\cdot$OPT. However, their result requires the hypothesis class $\mathcal{H}$ to be finite, whereas we only require finiteness of the strategic Littlestone dimension, which is a strictly weaker condition. We will add clarifications to revisions of our paper.
>
> > Question 2: Data points with no feasible manipulation that receives positive label would stay unmanipulated
>
> Yes, we do assume that data points that cannot manipulate to get a positive label would choose not to manipulate, and this is a necessary assumption for establishing our upper bound (whereas the lower bound would still hold if we remove this assumption). On the technical side, when false negative mistakes are made, this assumption enables us to identify the pre-manipulation features as the root of the strategic Littlestone subtree, and is crucial for establishing the “progress on mistakes” property. This assumption is also crucial for the previous upper bounds in [Ahmadi et al, Cohen et al.]. We believe that extending the results to the setting where all data points could manipulate is an interesting question for future research.
>
> > The importance of modeling the manipulation graph G accurately
>
> We completely agree that accurate modeling of the manipulation graph is crucial and that inaccurate estimations for certain graphs could lead to fairness issues. As a preliminary effort towards addressing unknown manipulation graphs, we allow the learner to run the strategic classification algorithm using knowledge about a potentially larger graph class that contains the true graph. The learner incurs only a logarithmic cost in the size of the graph class (assuming that taking union of the graphs does not significantly increase maximum in/out degree). Since this extra cost is only logarithmic, this (to some extent) allows the learner to adopt more conservative estimations of the manipulation that captures the types of manipulations from both groups. However, this approach focuses on the objective of minimizing the total number of mistakes rather than ensuring fairness among different groups. We believe the fairness implications that you described can be an interesting direction for future research.

---

> > ### Comment · Reviewer_wZa2 · 2024-08-12
> >
> > Thanks for your response.

---

### Decision · Program_Chairs · 2024-09-25

**Decision:**

Accept (poster)

**Comment:**

The reviewers are in agreement that this paper studies a well-motivated problem, and proposed a novel instance-dependent complexity notion (strategic Littlestone's dimension that depends on the hypothesis class and manipulation graph). Extensions to the agnostic setting and the unknown manipulation graph setting are deemed interesting. The rebuttal addressed many concerns of the reviewers (mainly from a clarity perspective). They are happy to recommend acceptance.